# ActionAtlas: A VideoQA Benchmark for Domain-specialized Action Recognition

**Mohammadreza Salehi**[◇]   **Jae Sung Park**[◇]   **Tanush Yadav**[◇]
**Aditya Kusupati**[◇]   **Ranjay Krishna**[◇†]   **Yejin Choi**[◇‡]   **Hannaneh Hajishirzi**[◇†]   **Ali Farhadi**[◇†]

[◇]University of Washington   [†]Allen Institute for AI   [‡]Nvidia
mrsalehi@cs.washington.edu
https://mrsalehi.github.io/action-atlas/

## Abstract

Our world is full of varied actions and moves across specialized domains that we, as humans, strive to identify and understand. Within any single domain, actions can often appear quite similar, making it challenging for deep models to distinguish them accurately. To evaluate the effectiveness of multimodal foundation models in helping us recognize such actions, we present ActionAtlas v1.0, a multiple-choice video question-answering benchmark featuring short videos across various sports. Each video in the dataset is paired with a question and four or five choices. The question pinpoints specific individuals, asking which choice "best" describes their action within a certain temporal context. Overall, the dataset includes 934 videos showcasing 580 unique actions across 56 sports, with a total of 1896 actions within choices. Unlike most existing video question answering benchmarks that only cover simplistic actions, often identifiable from a single frame, ActionAtlas focuses on intricate movements and rigorously tests the model's capability to discern subtle differences between moves that look similar within each domain. We evaluate open and proprietary foundation models on this benchmark, finding that the best model, GPT-4o, achieves a maximum accuracy of 45.52%. Meanwhile, Non-expert crowd workers, provided with action description for each choice, achieve 61.64% accuracy, where random chance is approximately 21%. Our findings with state-of-the-art models indicate that having a high frame sampling rate is important for accurately recognizing actions in ActionAtlas, a feature that some leading proprietary video models, such as Gemini, do not include in their default configurations.

## 1 Introduction

Multiple institutions have remarked on a striking finding: In many standard video benchmarks [59, 29, 77, 76, 74], a single frame without any modeling of temporal dynamics was enough to perform well [33, 38]. The phenomenon, known as static appearance bias [33], allowed models to easily discern the action depicted in the video. For example, simply recognizing presence of a pool was sufficient to identify the action as "diving" from a single frame [43, 69]. In reaction, later action classification datasets were designed to capture a richer distribution of temporal understanding [17, 58, 57] and model architectures also evolved to incorporate the now "necessary" temporal dynamics [40, 13, 12, 73]. However, such datasets and models still do not cover all the complexities of real-world video understanding and are only limited to traditional classification setups. The issue is even more pronounced in video-language tasks; with the rise of foundation Vision-Language models (VLMs), we find ourselves *back at square one*: Tasks such as video question answering [75, 37, 77] and video-language retrieval [77, 21, 28] can be solved easily yet again by training on just a single or sparsely sampled set of frames from videos [4, 33].

38th Conference on Neural Information Processing Systems (NeurIPS 2024) Track on Datasets and Benchmarks.

Recognizing activities in many real-world videos requires an accurate identification of subtle changes in movements, posture, and interaction with the environment. This is particularly evident in numerous domain-specific videos as demonstrated in Figure 1. Actions within a specific domain may appear identical in a single frame but become identifiable across multiple frames that capture the *sequence of movements*, as shown by cross through and body twist motion in the Rose move example, or the continuous rotations in the water in the Cartwheel move. In certain instances, actions are so subtle that they remain challenging to distinguish even when presenting multiple frames, as shown in the Stutter step and 360 pressure flip examples in skateboarding. Action recognition in real-world videos becomes even more complex with multiple actors present, each engaged in distinct or relevant and often overlapping actions. The video of the soccer game exemplifies this, with players from both teams simultaneously performing different actions. A robust video-language model must be able to *track* individuals and activities amidst such multifaceted and busy videos.

To investigate whether current Vision-Language Models (VLMs) can address these challenges, we present Action Atlas v1.0, a multiple choice video question answering (VideoQA) benchmark. This preliminary version of the benchmark focuses on sports, a domain characterized by intricate actions which can stress test models with the challenges identified above. Each video in the dataset is paired with a question and four or five choices. The questions pinpoint specific individuals within, asking which choice "best" describes their action within a certain temporal context. Overall, the dataset includes 934 videos showcasing 580 unique actions across 56 sports, with a total of 1896 actions within choices. The videos in this dataset have an average duration of 6.07 seconds and an average frame rate of 32.18 frames per second (FPS).

To collect ActionAtlas, we develop a novel pipeline. In contrast to prior work that sourced action names from a single website [11, 48], we rely on GPT4's vast domain-specific knowledge and compile a comprehensive list of actions within each domain by prompting the model. Having this list, we crawl videos about those actions on YouTube. To further filter the obtained search results, we rely on numerous automatic filtering tools and techniques, such as exact and soft lexical search, and CLIP [53] filtering. Additionally, we show how LLMs and speech transcriptions can be used to faster find segments containing a specific action within long videos. To further ensure high quality of our benchmark, we incorporate multiple rounds of manual filtering carried out by both crowd-workers and the authors, who spent a month familiarizing themselves with the actions.

We evaluate open and proprietary VLMs, such as GPT-4o [50] and Gemini-Pro [63] on ActionAtlas. For all models except for Gemini models in video mode–which directly take input video files–, we follow the standard methodology from previous work [4]; we uniformly sample frames and feed them as image inputs concatenated with the question and the choices. For Gemini, we show how one can easily recover the exact frames that the model samples in video mode which makes it not much different from other VLMs. The random chance accuracy on our dataset is $20.91\%$, whereas the best open-weight model, Qwen2-VL-7B, performs only $\sim 30.24\%$ accurately. Meanwhile, GPT-4o reaches up to $45.52\%$. our results with both of these models show that increasing frame sampling rate helps significantly in recognizing actions in ActionAtlas, a feature that current top proprietary video models like Gemini lack in their default settings. Moreover, we show that providing action descriptions does not significantly improve model performance, while non-expert crowd workers achieve an overall accuracy of $61.6\%$ with those descriptions. This underscores the gap between AI models and human's visual recognition capabilities when it comes to actions with fine motions.

Taken together, our benchmark provides a new testing ground to evaluate foundation models on action recognition within specialized domains that have practical, real-world applications. The results on our benchmarks showcases that despite the improvements in image-understanding and long-form video understanding, true video-understanding is still lacking. Moreover, as demonstrated by previous work like Dall-E 3 [3], captions generated by AI models that understand nuanced moves–such as those in ActionAtlas–can enhance training of video generation models to better capture such nuances. Therefore, we believe ActionAtlas can help accelerate various aspects of video research.

## 2 Related work

### 2.1 Action Recognition

The seminal work by Schuldt et al. [56], which collected data on six basic human actions, set the stage for many subsequent work focused on action recognition, such as UCF101[60], HMDB51[68],

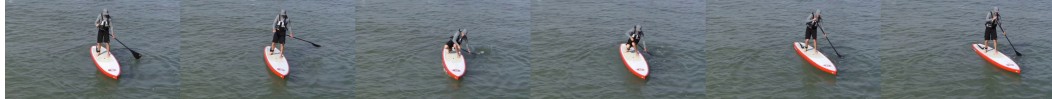

What best describes the move made by the player wearing a black jersey with number 10 after their teammate wearing number 13 passes the ball?

1. Bicycle kick    2. Rainbow flick    3. Elastico    4. Cruyff Turn    5 . Dummy ✅

What best describes the move made by the only player wearing a basketball jersey after they start running towards the basket and before making a jump shot?

1. Pull-up jump shot  2. Behind-the-back dribble  3. Stutter step ✅  4. In-and-out dribble  5. Crossver dribble

What best describes the move made by the man in the kayak?

1. Bow stall    2. Cartwheel ✅    3. Space Godzilla    4. Mcnasty    5. Blunt

What best describes the move made by the person?

1. Rose move ✅    2. Toe hook    3. Mantle    4. Gaston    5. Heel hook

What best describes the move made by the skateboarder?

1. Inward heelflip    2. Hard flip    3. Laser flip    4. 360 flip    5. 360 pressure flip ✅

What best describes the action performed by the man in black shorts?

1. Bracing ✅    2. Reverse stroke    3. Side slip    4. Forward stroke    5. Nose draw

Figure 1: **Examples from ActionAtlas**. To answer ActionAtlas's questions, models have to be able to recognize fine movements and nuances that differentiate actions belonging to the same domain (examples 2, 3, 4, 5, 6 from top), correctly localize and track the individual performing the action if there are many (example 1). [Video links from top to bottom: 1, 2, 3, 4, 5, 6].

ActivityNet [11], AVA [19], Kinetics [25], and Moments in time [48]. These studies have primarily focused on coarse-grained everyday human actions that are sourced from a single website [67] and are relatively easy for image models to recognize [53]. The Something-something v2 [17] benchmark has also introduced 174 diagnostic commonsense actions to assess models' understanding of world physics. Breakfast [30] and MPII Cooking [55] datasets have collected 10 and 67 fine-grained actions in cooking, and Diving48 [38] and Finegym [57] collected fine-grained actions in only diving and gymnastics respectively with just a single action actor in each video. In domains with many

individuals in the video, Multisports [39] is the biggest dataset which spans 67 fine-grained actions in four sports. Other work [79, 42, 78, 39, 20, 15, 31, 46, 88] have created datasets for fine-grained action recognition in different sports such as soccer, basketball, figure skating, diving, fencing, table tennis, etc. Our work differs from all these work in the following aspects: 1. **Larger number of domain-specialized actions**. There are 580 ground truth actions depicted in videos in ActionAtlas. Moreover, as there are 1896 total actions in choices, the dataset effectively tests the model's capability to discern 1896 actions which is more than 3 times larger than previous datasets, such as Finegym. 2. **The actions in ActionAtlas represent knowledge-driven, real-world actions**, In contrast to datasets like Something-something v2 [17], our work focuses on evaluating foundation models on action recognition within specialized domains that have practical real-world applications. This is analogous to the recent trend on evaluating foundation models on specialized domains with datasets like MMLU [22] and MMMU [83] in the text and image space. 3. **The use of language and QA in ActionAtlas**. Previous work [39] have used bounding boxes to refer to individuals engaged in a particular action in a video with many people in it. However, we follow works like refCOCO [81] and use natural language to refer to individuals (see how questions refer to action actors in Figure 1). This aligns more closely with how humans naturally refer to an object or person and also better fits evaluating VLMs. 4. **Faster discovery of actions using LLMs**. We show how the extensive knowledge of LLMs can be used to identify a broad range of actions, including rarer actions that experts may overlook. Furthermore, we show how LLMs such as GPT4-text can be used along with speech transcription to find candidate segments within longer videos that are likely to contain a specific action (see §3.4 and Figure 3). This approach relies solely on text and can help making the collection pipeline more scalable.

## 2.2 Video QA

Most Video-QA datasets, such as NextQA [74], ActivityNet-QA[82], VATEX [71], MSRVTT-QA and MSVD-QA[76] are designed for general purpose video understanding, with most questions about visual appearance of the scenes or basic actions in the video which fall under the category of common sense understanding. These do not pose a significant challenge to current state-of-the-art VLMs. Other datasets like MVBench [36] and Video-Bench [49] have attempted to standardize existing datasets into a multiple-choice QA format but still primarily cover simplistic actions. Ego4D [18] and Ego-Exo4D [9] have annotated various actions performed by crowd workers, but these do not include many expert-level actions in the domains they cover as performing such actions is extremely hard for non-expert crowd-workers. While Youcook2 [87] contains densely annotated cooking videos, most of the actions (e.g., pouring water into a cup) are extremely simple with more expert-level actions overlooked as the collection pipeline is not designed for capturing them. Additionally, Perception test [52] is another diagnostic benchmark for evaluating basic world understanding of models. In contrast, ActionAtlas's focus is on real-world actions in specialized domains.

## 2.3 Long-form Video Understanding

Video, as a form of streaming data, can get infinitely long, making it challenging for deep models to understand them. Recently, there has been growing interest in creating long-form video understanding datasets from various sources. Several studies have used long movies to curate such datasets [62, 32, 2]. A common limitation is that models can often answer the questions based on just the story-line–which they might have seen in their pretraining data–without the need to attend to the visual content. Other works such as EgoSchema [45] have used crowd-sourced Ego4D videos to create QA datasets with long (180 seconds) ego-centric videos; however, the actions in these datasets tend to be coarse-grained and simplistic. Furthermore, as noted in prior work [47, 63], it is still uncertain whether viewing the full video is necessary to answer the questions. This uncertainty arises because the questions generated by LLMs are not carefully filtered for language biases which helps models answer solely based on text. In contrast to studies focused on long-form video understanding, ActionAtlas specifically targets short videos about domain-specific actions, many of which characterized by rapid changes and movements within the scene. Furthermore, recent studies such as [84, 47] have shown that it is possible to use a video captioner to summarize smaller chunks of a long video and pass them to an LLM for reasoning over the entire video. Therefore, understanding short video clips and the potentially complex actions they depict could serve as a foundation for solving long-form video understanding.

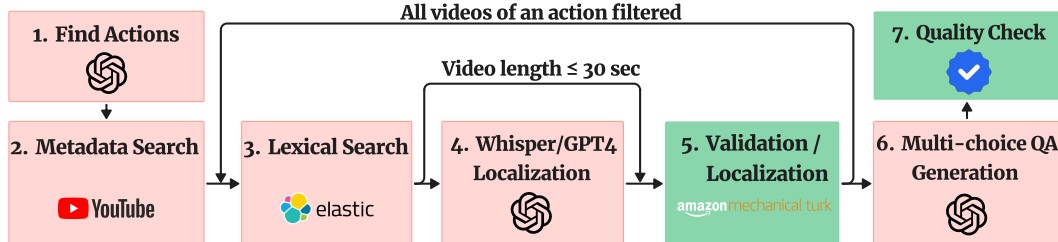

Figure 2: **Data collection pipeline consisting of** Automatic **and** Manual **parts.** First a comprehensive list of actions is compiled (§3.1) which are then used for searching metadata of videos on YouTube relevant to each action (§3.2). Then with lexical search a subset of videos are selected (§3.3). If a video is shorter than 30 seconds, it will be used in crowd-sourcing. Otherwise, the video is transcribed, and GPT4 selects potential 30 second segments that contain the actions based on the transcription (§3.4). Mechanical Turkers will then verify the presence of actions in the segments (§3.5) and localize it (§3.6). If all videos of an action were rejected, we repeat the process to source new videos. Finally, GPT4 generates Multiple-choice QAs which are checked by the authors (§3.7).

## 3 Collection

Our goal is to create a high quality benchmark of actions within specialized domains. Since sports encompass a wide array of such actions, in ActionAtlas v1.0 we focus on actions within various sports. Nevertheless, we would like the collection process to be scalable, allowing expansion to many more domains in the future. Toward this, we develop a robust pipeline that incorporates automatic filtering using various tools and AI models, followed by multiple rounds of manual filtering by crowd-workers and the authors. Figure 2 illustrates the pipeline for collecting ActionAtlas. As we move forward to the latter stages of this pipeline, the need for manual verification and expertise increases, making it more and more expensive. In the final and most costly stage, the goal is for trained annotators to only verify the already vetted annotations from previous stages and refine them if needed–instead of asking them to directly search for videos on YouTube. Our data collection also relies on one fundamental assumption: By starting from a very large initial set of videos for each action, we ensure that even after an intensive automatic filtering, a large enough number of videos remain for humans to check. Note that whenever we mention "GPT4" or "GPT4-text" in our pipeline it specifically refers to `gpt4-1106-preview` *without using vision capability*.

### 3.1 Compiling A List of Actions

To generate the action list, we first collected the names of 150 most popular sports through prompting GPT4. To gather the actions within each sport, previous work [39, 79] asked experts to write the name of the moves. However, we witnessed that those lists were incomplete as humans tend to overlook actions in the far tail of the distribution. We therefore decided to ask GPT4 to list the moves for us. We found that using GPT4 with a prompting strategy which we call *expansion and squeeze prompting* resulted in the best coverage. In this approach, GPT4 is prompted to generate an action list for any domain given few-shot examples in an example domain (e.g., golf). As we found that the model may still omit some major actions, we iteratively expanded this list by using the model's previous outputs as new few-shot examples. After two rounds of expansion, we squeezed the list by having GPT4 identify and remove false positives, i.e. phrases that were added during expansion and considered to be physical actions but in reality are not. In total, The expansion process resulted in 19.5k actions which were reduced to 10.5k after squeezing. It is worth noting that before solely relying on GPT4 to get the action list, we tried to compile a list by crawling data from available knowledge-bases, such as DBPedia and Wikipedia. However, we found that many actions are missing in those knowledge bases. Please refer to Appendix F for GPT4 prompts used in this section.

### 3.2 Searching for Metadata of Videos

Searching for videos of a large list of actions and downloading both the audio and video streams of the results would be impractical due to the high storage and computational costs involved. Instead, we initially queried and downloaded metadata of videos on YouTube, which includes the titles,

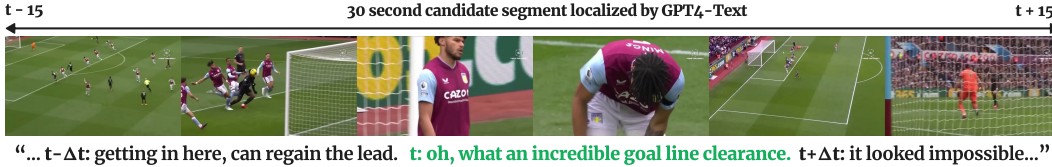

"... t−Δt: getting in here, can regain the lead.    t: oh, what an incredible goal line clearance.    t+Δt: it looked impossible..."

t' − 15                                                                          t' + 15

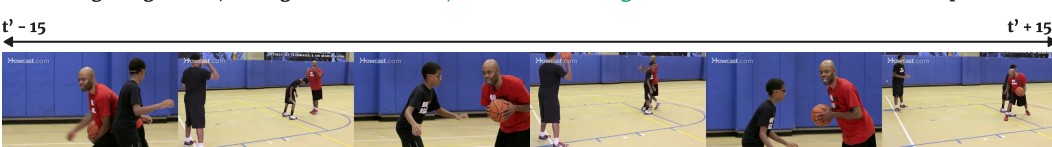

"... t'−Δt: I wanna wrap the ball around them where now I'm feeding my teammate. t': Ok, so I'll do that one more time.
t'+Δt: The defense is here, and as I'm dribbling the basketball or I have it in triple threat, I'm gonna come from this
motion, I'm handling the ball step around and pass..."

Figure 3: Given transcription of a long video, GPT4-Text can be prompted to output timestamps where the action is likely to occur, *without having access to frames*. The model has found instances where the speaker comments on the great quality of the action (top) or indicates that a demonstration of the action is going to happen shortly (bottom). More details in §3.4. [Video links: top, bottom].

descriptions, and subtitles. This enabled us to carry out text-based filtering as our next step to eliminate any irrelevant videos from the search results, leaving us with videos that are highly likely to contain the desired actions which will then be downloaded. We formed queries by concatenating the name of action to its domain (e.g. "knuckleball shot soccer") and searched for YouTube metadata for each query. This yielded a total of approximately 4.5 million unique and valid metadata files, or about 450 valid metadata files per action. We will describe the text-based filtering process next.

## 3.3 Lexical Search

Our extensive metadata search for each action yielded a high recall of YouTube videos associated with that action. However, we also encountered false positives in our results–videos not sufficiently relevant to the specific action and domain of our search. To narrow down our selection to videos more likely to contain the targeted action, we used lexical matching on video titles via Elasticsearch [8]. Specifically, we searched for videos with the action name appearing exactly within the title. Additionally, we restricted our search to videos with more than 1000 views as of the collection date. Moreover, the BM25 engine in Elasticsearch provided a soft search feature, enabling us to detect whether video titles included synonyms of the action query (e.g., "knuckleball shot" and "knuckleball kick" in soccer) that exact matching might have missed. In the lexical search process, we limited the number of hits to 100 videos per each action.

## 3.4 Finding Potential Segments in Long Videos via Whisper and GPT4-text

The primary goal of the automatic filtering pipeline is to identify videos that are highly likely to contain the desired actions. These videos are subsequently forwarded to crowd workers to confirm the presence of these actions. Many high quality action demonstrations happen in long videos, specially in the form of tutorials. As watching long videos can be tedious and costly in terms of annotation, we decided to extract potential 30-second clips that are highly likely to contain the target actions within longer videos. Videos shorter than 30 seconds directly proceeded to the validation stage done by crowd-workers. To extract potential segments, we focused on videos with speech data and first transcribed them using OpenAI's Whisper model [54]. The transcripts, which included timestamps provided by Whisper, as well as the action name, served as context for GPT4. The model was then prompted to output timestamps where the action is most likely to occur, focusing specifically on instances where the speaker comments on the great quality of the action (Figure 3 top) or indicates that a demonstration of the action is going to happen shortly (Figure 3 bottom). We then extracted 15 seconds before and after the suggested timestamp as the candidate segment. These segments were further filtered using CLIP similarity score [53] with their corresponding domain name and video domain pairs with a cosine similarity of lower than 0.1 were discarded. In total, we transcribed and localized 57k videos with Whisper and GPT4-text, and collected 49k videos that were shorter than 30

**Half-Volley Drop Shot:** In tennis, a half-volley drop shot is a delicate and skillful maneuver that involves the player hitting the tennis ball just after it has bounced, without allowing it to rise to its full height. The player aims to barely touch the ball with a gentle tap, causing it to cross just over the net and land softly on the other side, making it difficult for the opponent to reach and return.

**Simple Steps to Identify a Half-Volley Drop Shot:**

**1. Identify the Player Hitting the Ball:** Focus on the player who is about to hit the tennis ball or has just hit it. It's important because you need to watch their movements closely.

**2. Look at the Ball Bounce:** Wait for the moment when the ball bounces on the side of the player you're watching. A half-volley drop shot is executed right after the bounce.

**3. Observe the Stroke:** Pay attention to how the player hits the ball. In a half-volley drop shot, the racket will gently tap the ball near the ground level instead of swinging through it. The action is subtle and controlled.

**4. Watch the Ball's Trajectory:** The key to identifying a half-volley drop shot is the trajectory of the ball after it's hit. The ball should barely clear the net and then fall quickly to the ground, with minimal bounce on the opponent's side.

Figure 4: **Definition of Half-Volley Drop Shot generated by GPT4-text to be used by crowd-workers for validation and localization.** The workers match the key elements listed in the definition with what they see in the video to identify if the action happens. For more details see §3.5.

seconds and did not need localization. All these videos are downloaded at end of this stage. Please refer to Appendix F for the prompts used with GPT4.

## 3.5   Validation via Crowd-sourcing

After collecting candidate 30 second segments likely to contain actions, we further validated the presence of actions with the help of crowd-workers on Amazon Mechanical Turk. To do so, we sorted the actions based on the number of videos filtered from previous stages and performed the validation process in batches, starting from the actions with highest number of videos and proceeding in descending order. A potential challenge here was the workers' unfamiliarity with actions in different domains which might introduce errors in the annotation. To mitigate such errors, we employed multiple safeguards in our pipeline: 1. We prompted GPT4-text to provide descriptions that highlight the key elements required to recognize an action. An example definition is shown in Figure 4. 2. We presented five potential 30 second segments per each action to workers instead of one. This allowed the worker to compare the videos against each other and against the description which further helps in identifying the action. Workers were instructed to watch each video and determine if the action happens or not. If the worker noticed any discrepancy between the videos and the description provided by GPT4, they could search on video platforms to watch more videos and educate themselves on the action. 3. Each set of videos was reviewed by three workers, and only videos for which at least two workers confirmed the presence of the action proceeded to the next step. If all videos of an action were rejected by crowd-workers, we repeated the earlier stages of the pipeline to source new videos for that action (see the arrow from step 5 to step 3 in Figure 2). For more details on verification via crowd-sourcing and Mechanical Turk layout see Appendix K.

## 3.6   Localization via Crowd-sourcing

As many events and actions can happen in a 30 second video clip, we sought to find a smaller segment that better isolates the target action. However, isolating a single action in a video clip can be quite challenging. This complexity arises because some actions last only briefly (e.g., stutter step, as shown in Figure 1), while some are only defined based on what happens before or after them (e.g., a dummy, as shown in Figure 1). To address these challenges, we asked crowd workers to specify three key details if necessary: 1. Some attributes that uniquely identify the action actor; they were instructed to use any distinguishing feature (e.g., number or name on jersey, clothing or hair color), as long as it uniquely identifies the individual(s) performing the action. 2. The events that occur immediately before and after the action. We then asked them to propose a start and end time stamp within the 30 second segment that encapsulates these two pieces of information. This data were then used to extract the final video segments in the benchmark and write questions about them. For more details on localization via crowd-sourcing and Mechanical Turk layout see Appendix K.

### 3.7 Multiple-choice QA Generation and Quality Check

We prompted GPT4-text with specific question templates and the two pieces of information obtained in §3.6 and asked it to write a question about what the action is (see Figure 1 for example questions). We also prompted the model to write 9 hard negatives for each action. Subsequently, for the quality check, a team of three individuals including two of the authors, trained themselves on recognizing the actions and checked the questions, videos, and the hard negatives over the span of a month. For the hard negatives, we kept only the three or four most plausible ones. Videos for which we were uncertain about the presence of the ground truth action were discarded. To eliminate questions answerable solely by text, we passed them through GPT4-Text model three times. We noticed that in 9% of the samples the model consistently predicted the correct answer. This was often due to the questions inadvertently revealing information about the answers. To address this, we rewrote the questions and choices. Moreover, to manage instances where text within video frames might leak information about the ground truth action, we used the Google Cloud Vision API [1] to detect such text, and blurred it using Gaussian noise. Appendix F provides more details on the prompts used in this section.

## 4 Evaluation

### 4.1 Models and Baselines

**CLIP.** We evaluated OpenAI's CLIP ViT-L-14-336 [53] as a foundation VLM without any large language models in it. To form the prompts, we asked GPT4-Text to rewrite each question as a sentence and embed each choice into the sentence. This gave us four or five plausible class prompts which we used to do classification with the model.

**Proprietary large VLMs.** We evaluated multiple proprietary VLMs on ActionAtlas: Gemini 1.5 Flash and Gemini 1.5 Pro using the Gen AI API [16] (model versions `gemini-1.5-flash-latest` and `gemini-1.5-pro-latest`), GPT4-o [50], GPT-4 Turbo, and GPT-4o-mini using the Open AI API[1]. The GPT-4 family cannot directly process video inputs. Therefore, we uniformly sampled different numbers of frames–1, 4, 8, 16, and 32–and reported the results for each configuration. For Gemini models, as detailed in §4.3, we discovered that the video mode always samples a specific set of frames at a rate of one FPS, making it not much different from other models. With such sampling rate, to make the model use all the frames in a video as input, we converted our videos to 1 FPS videos (e.g., a 3-second, 30 FPS video would become a 90-second, 1 FPS video) and evaluated the Pro variant on these converted videos as well.

**Open large VLMs.** We evaluated the following open models on ActionAtlas with uniformly sampled frames: Qwen2-VL-7B [70], mPLUG-Owl [80], Video-LLaMA [85], Video-LaVIT [24], VideoChat v2 [36], and LLaVA-Next-Video [86]. Frames were down-scaled to $336 \times 336$ for models whose image encoders supported this resolution; If not, frames were scaled to the maximum supported resolution (e.g., $224 \times 224$). For Qwen2-VL-7B, we down-scaled the frames while maintaining the aspect ratio so that each frame does not have more than $336 * 336$ pixels in total. All evaluations with open models were done on a single H100 GPU. Further details about the setup can be found in Appendix A.

**Non-expert humans.** To get non-expert human's performance as a baseline, we asked crowd-workers on Amazon Mechanical Turk to respond to ActionAtlas's questions. As the workers might not be familiar with the action names, we generated two to three sentence descriptions of each action choice using GPT-4o, which were then provided to the workers along with the available choices. The workers were asked to answer the questions without using YouTube, but they were allowed to perform text search on Google or use AI chatbots, provided the chatbots' vision capabilities were not used. Note that in the ablation experiments described in §4.3, we also evaluated AI models when these descriptions are provided.

---

[1]All the Gemini and OpenAI models were their latest versions as of July 31, 2024.

Table 1: **Evaluation results on ActionAtlas for open models.** Open models perform no better than random chance. The efficiency metrics are averaged across the benchmark. *a single video frame and 24 frames of motion vectors are used, consistent with the video tokenizer described in [24].

| Model | #Input frames | #Input video tokens | #Inference TFLOPs | Accuracy(%) |
|---|---|---|---|---|
| Random chance | - | - | - | 20.91 ±2.49 |
| Non-expert human | - | - | - | 61.64 ±3.29 |
| CLIP ViT-L-14-336 [53] | 16 | 16 × 576 | - | 23.71 ±2.62 |
| mPLUG-Owl-video [80] | 16 | 16 × 256 | 2.94 | 19.49 ±2.68 |
| Video-LLaMA [85] | 16 | 16 × 256 | 6.12 | 22.71 ±2.69 |
| Video-LaVIT [24] | 24* | 24 × 135 | 3.38 | 19.37 ±2.46 |
| VideoChat2 [36] | 16 | 16 × 196 | 3.23 | 20.86 ±2.34 |
| | 32 | 32 × 196 | 4.69 | 20.77 ±2.67 |
| | 64 | 64 × 196 | 7.51 | 21.27 ±2.75 |
| LLaVA-Next-Video-7B [86] | 16 | 16 × 144 | 22.0 | 20.77 ±2.67 |
| | 32 | 32 × 144 | 43.0 | 21.06 ±2.64 |
| | 64 | 64 × 144 | 83.2 | 22.90 ±2.56 |
| Qwen2-VL-7B [64] | 2 | 1 × 576 | 2.45 | 24.07 ±2.66 |
| | 4 | 2 × 576 | 4.31 | 26.71 ±2.83 |
| | 8 | 4 × 576 | 8.31 | 27.75 ±2.81 |
| | 16 | 8 × 576 | 13.38 | 30.24 ±2.94 |
| | 32 | 16 × 576 | 29.87 | 29.33 ±3.10 |

## 4.2 Results

We benchmark all models using only the video data, discarding any audio, transcriptions, or captions. For open models, frames are uniformly sampled throughout the video and the models are evaluated based on the following metrics: the number of input frames sampled by the model, the number of tokens to which the video is compressed before being fed to any transformer model, average inference FLOPs on the benchmark (calculated using fvcore [26] package), and top-1 accuracy (for more information on the significance of these metrics, see §5). Following [10], we report 95% confidence intervals via bootstrap sampling. For the results based on sampling fixed number of frames per second see Appendix B.

The results in Table 1 indicate that, except for Qwen2-VL-7B, open models often perform no better than, or only slightly better than random chance. This suggests that open models struggle to capture the nuances of complex actions in specialized domains, likely due to the absence of such data in their pretraining corpora. For Qwen2-VL-7B, increasing the number of frames from 2 to 16 increases accuracy statistically significantly. However, the performance declines

Table 2: **Evaluation results for proprietary models.**. For Gemini in video mode we either provide the original video or transform it into a video with 1fps and then evaluate the model. See §4.3 for more details on how Gemini processes inputs.

| Model | #Input frames | Accuracy(%) |
|---|---|---|
| Gemini 1.5 Flash (video) | 1 fps | 30.49 ±2.86 |
| Gemini 1.5 Pro (video) | 1 fps | 32.37 ±3.04 |
| Gemini 1.5 Pro (video) | all frames | 35.59 ±3.04 |
| GPT-4 Turbo | 1 | 28.97 ±3.06 |
| | 4 | 33.17 ±2.98 |
| | 8 | 34.25 ±3.09 |
| | 16 | 33.99 ±3.05 |
| | 32 | 32.24 ±2.80 |
| GPT-4o-mini | 1 | 30.15 ±2.74 |
| | 4 | 33.42 ±2.70 |
| | 8 | 30.20 ±2.89 |
| | 16 | 32.20 ±2.76 |
| | 32 | 31.17 ±2.90 |
| GPT-4o | 1 | 33.08 ±2.89 |
| | 4 | 39.50 ±3.10 |
| | 8 | 41.55 ±3.01 |
| | 16 | **42.95 ±2.91** |
| | 32 | 41.44 ±3.11 |

or plateaus upon reaching 32 frames. This indicates that while sampling more frames provides valuable information for understanding complex domain-specialized actions, there is also a downside of handling longer video sequences that open video models might struggle with. Regarding number

Table 3: **The accuracy improvements from providing action choices' descriptions is not statistically significant.**

| Model | Acc. without Description (%) | Acc. with Description (%) |
|---|---|---|
| mPlug-owl | 19.49 ±2.68 | 20.92 ±2.51 |
| Video-Llama | 22.71 ±2.69 | 21.85 ±2.68 |
| LLaVA-Next-video | 20.77 ±2.67 | 22.20 ±2.50 |
| Qwen2-VL | 30.24 ±2.94 | 33.13 ±3.04 |
| Gemini Pro 1.5 | 32.37 ±3.04 | 37.00 ±3.06 |
| GPT-4o | 42.95 ±2.91 | 44.05 ±2.94 |

of video tokens and inference FLOPs, for all models except for Qwen2-VL-7B, we do not see a straightforward relationship with the performance, suggesting that increasing computational complexity does not necessarily lead to better results for these models. However, for Qwen2-VL-7B, there is a notable positive correlation up to 16 frames. Additionally, Qwen2-VL-7B compresses frames into fewer tokens compared to models processing them at the same spatial and temporal resolution. For instance, with 16 frames, Qwen2-VL-7B uses half the tokens that CLIP uses, while delivering significantly better performance.

For the GPT-4 family, increasing the number of sampled frames does not lead to statistically significant improvements for GPT-4 Turbo and GPT-4o-mini. However, for GPT-4o, sampling more frames results in statistically significant improvements, rising from 33.08% to 42.95%. For video mode of the Gemini family, both Pro and Flash variants achieve comparable accuracies given the confidence intervals. Furthermore, there is no statistically significant improvement when increasing the number of frames from default 1 fps to all video frames (see §4.1 for how we evaluate in that mode) with the Gemini 1.5 Pro model. Nonetheless, as discussed in the ablation studies in §4.3, sampling a single frame from the video results in the Pro model achieving an accuracy of 28.73%, which is significantly lower than the 35.59% when using all frames of the video.

## 4.3 Ablations

**Providing description of actions.** Correctly answering the questions in ActionAtlas requires two capabilities: 1. Having knowledge about the action choices, specifically recognizing the action names, and 2. Visually recognize the action in the video. To show that model failures are primarily due to poor visual recognition, we provided descriptions of the action choices to the models. These descriptions were created by prompting GPT-4o to summarize the key elements needed to recognize each action in 2-3 sentences. As shown in Table 3, adding these descriptions did not yield statistically significant improvements on our dataset. However, using the same descriptions, humans outperformed the best proprietary model, GPT-4o, by 18%. This suggests that the model struggles on our dataset not due to a lack of knowledge about the action names, but because of limitations in visual recognition.

**Chain-of-thought reasoning.** Chain-of-thought (CoT) reasoning or generating intermediate reasoning steps [72, 27] has shown to be effective in improving performance across different language and vision-and-language tasks [7, 22, 83]. To test this on our dataset, we instruct models to reason step by step and provide their rationale when making a choice. We choose not to include few-shot examples to avoid occupying a significant portion of the models' available context length. Table 4 shows the results when using CoT reasoning with or without choice descriptions. Surprisingly, for all models there is a significant drop in performance when using only CoT. Although adding choice descriptions improves performance in the case of GPT-4o, the improvements are not statistically significant compared to the original results. This suggests that enhanced reasoning alone does not improve performance on ActionAtlas and the low performance of models is mainly due to poor visual recognition. We also experimented with prompting models to reason step by step across frames, by describing differences between each pair of consecutive frames, as done in previous studies [5]. Yet, this strategy does not enhance performance. It is also worth noting that with Gemini Pro 1.5, there is a dramatic increase in the number of refusals, which might contribute to the drop in performance; Table 8 in Appendix E shows that with the chain-of-thought setup, the refusal rate of Gemini 1.5 Pro model increases to more than 5%.

Table 4: **Improvements from Chain-of-thought reasoning are not statistically significant on ActionAtlas.** See §4.3 fore more details on the setup.

| Model | Acc. (%) |
|---|---|
| GPT-4o | $42.95 _{\pm 2.91}$ |
| + CoT | $36.24 _{\pm 3.05}$ |
| + CoT + Choice Description | $45.52 _{\pm 3.02}$ |
| Qwen2-VL | $32.49 _{\pm 3.05}$ |
| + CoT | $26.23 _{\pm 2.79}$ |
| + CoT + Choice Description | $26.54 _{\pm 2.88}$ |
| Gemini Pro 1.5 | $32.37 _{\pm 3.04}$ |
| + CoT | $24.36 _{\pm 2.73}$ |
| + CoT + Choice Description | $27.68 _{\pm 2.84}$ |

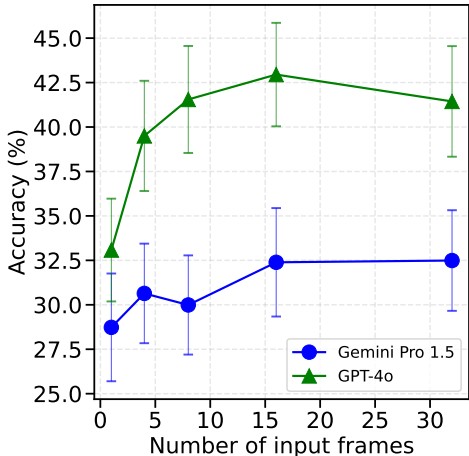

Figure 5: **Effect of changing frame sampling rate for GPT-4o and Gemini Pro 1.5 models on accuracy on ActionAtlas**. See §4.3 for more details.

**Changing frame sampling rate with Gemini Pro 1.5** In our evaluation of proprietary API models, the Gemini family were the only ones at the time that could directly take video input files[2]. While investigating how these models process videos by inserting random noise images at different frame positions, we noticed that the these models *always and only* process the middle frame in each second of a video, resulting in a processing rate of 1 FPS. This finding aligns with the Gemini technical report [63], wherein all the evaluations are done at 1 fps. Processing a preselected set of frames, not only exposes the model to potential frame injection attacks but also limit its ability to detect motions occurring more frequently than this sampling rate (for more details on the frame injection attack and jail-breaking Multi-modal Gemini models, see Appendix D). To test the model at a higher sampling rate in video mode, we converted all the videos in ActionAtlas to 1 FPS videos–meaning each video has only one frame per second–and re-evaluated the model. As shown in Table 2 and discussed in §4, no significant improvements were observed. This suggests that Gemini models might not be trained to handle all the frames within a video. Furthermore, For a better comparison with other models, including GPT-4o, we also sampled a fixed number of frames from the videos and evaluated Gemini Pro 1.5 in image mode. Figure 5 indicates that with Gemini 1.5 Pro, increasing the number of sampled frames does not yield as substantial improvements as it did with GPT-4o. It is worth mentioning that as Table 8 in Appendix E shows, we noticed a much higher refusal rate with the models when input frames were used (from $1.39\%$ to $5.14\%$), similar to the findings from the CoT experiments in §4.3. This refusal rate also slightly increased with the addition of more frames.

## 4.4 Qualitative Error Analysis

To understand why VLMs struggle on our benchmark, we investigate the nature of the errors made by the Gemini and GPT-4 family. We sample 20 erroneous test cases at random, and analyzed models' reasoning. We conduct this analysis within two setups: one where descriptions of the choices are provided, and one where only the action names are provided. We prompt the model to use chain-of-thought reasoning in both setups.

We find that most errors fall into *at least* one of two broad categories and four subcategories. Namely, visual hallucinations, visual oversights, and visual tracking failure are subcategories within the visual recognition errors category, while the remaining errors are classified under the QA reasoning failure category. Figure 6 shows examples of these errors.

---

[2]We noticed ChatGPT web app could directly take video input files. However, the model reports that it extracts key frames from the video via tools such as a Python program and packages like OpenCV and Matplotlib. Furthermore, we found the resulting output to be of poor quality and sometimes unrelated to the original video content.

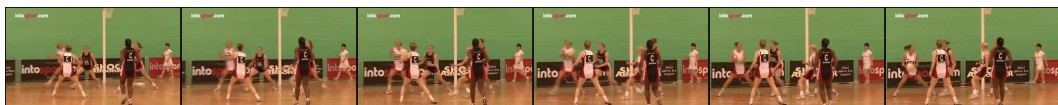

(a) **Visual hallucination in netball.** The clip shows a chest pass, with the player with a black-and-white C on their jersey holding the ball at their chest and using both hands to throw it to their teammate. GPT-4o mistakenly thinks that the "the pass is directed downwards to the floor before it reaches the teammate," but it is clear from the clip that the pass is directed straight to the teammate. This video failure causes GPT-4o to make the incorrect prediction of "bounce pass." [Video link]

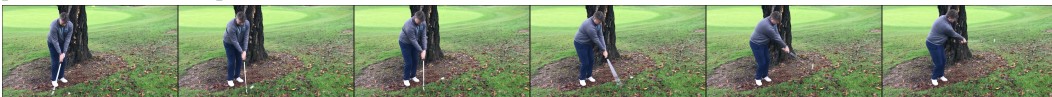

(b) **Visual oversight in golf.** This trimmed clip shows a double hit, where the first hit is a chip shot. Both GPT-4o and Gemini Pro 1.5 identify chip shot as the option that "best describes the action" in the clip. However, double hit is the more favorable option as the way the golfer hits the ball twice is illegal. Both GPT-4o and Gemini Pro 1.5 focus purely on the initial swing and overlook what happens after it (namely, that the club hits the ball again), causing the incorrect prediction. [Video link]

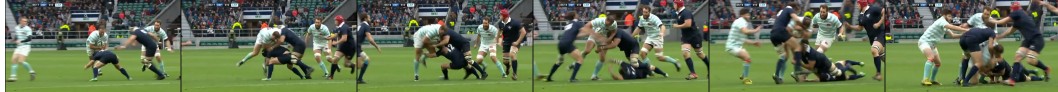

(c) **QA reasoning failure in rugby.** This trimmed clip shows a crash ball, where the player charges directly at the defense. GPT-4o concludes that "this move is characteristic of a crash ball," but confusingly outputs "offload pass" as its prediction. It is unclear why a model would arrive at one option in its chain-of-thought, and then decide on another option for its final answer. [Video link]

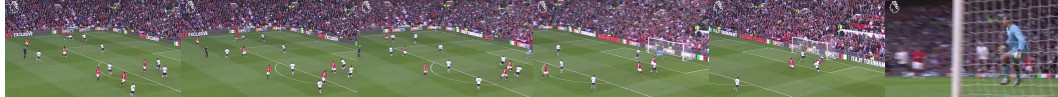

(d) **Visual tracking failure in soccer.** The clip shows a player crossing the ball and player #7 performing a diving header. GPT-4o concludes that "subsequent frames show player #7 crossing the ball into the penalty area towards the goal" and chooses "cross" as the answer. This shows that the model has failed to track player #7 and confused it with the other player. [Video link]

Figure 6: **Examples of prediction errors by proprietary models on ActionAtlas.**

Visual hallucinations occur when a model misidentifies an item or action in a video clip and hallucinates another action. For instance, in the netball example shown in Figure 6a, GPT-4o hallucinates that the ball bounces on the ground during a pass, whereas the video clearly shows a direct chest pass. Visual oversights happen when a model correctly identifies most elements of a clip but misses a crucial detail or movement in the clip. An example is the golf clip of a double hit shown in Figure 6b, where the first of the two hits is a chip shot. Both GPT-4o and Gemini 1.5 Pro focus on the swing but overlook the obvious second hit. This causes the models to incorrectly predict chip shot, which is not the "best" choice that describes the video as asked by the question. Visual tracking happens when the model fails to localize and track the individual performing the action which is depicted in 6d. Lastly, QA reasoning failures, shown in Figure 6c happen when the model describes the correct action but ultimately selects the wrong choice. Overall, visual hallucinations are the most common error across-the-board, making up about $60\%$ of errors. The remaining errors are divided among visual oversights, tracking and reasoning failures, with visual oversights being slightly more prevalent.

## 5 Discussion

**Why do the reported metrics matter?** The quantitative and qualitative results with models like GPT-4o have shown that accurately detecting subtle movements in actions within ActionAtlas requires denser video signal sampling–which in its simplest form means sampling more frames. While a higher sampling rate could lead to a greater number of tokens and thus higher FLOPs if tokenized naively, much of the additional data from increased sampling is often redundant. Better tokenizers could potentially leverage this redundancy and compress the sampled data to maintain the same number

of tokens as would be achieved with a lower frame sampling rate, without sacrificing downstream performance. We need further research on tokenizers to find the optimal balance between video sampling rates and token compression, while ensuring high downstream accuracy on vision-and-language tasks. Simple strategies such as masking spatiotemporal patches [14], or more innovative tokenization schemes that go beyond mere frame sampling, could help maintain a stable token count even with higher sampling rates. An example of such progress is Video-LaVIT [24], which encodes frames following a key frame into more compact representations using motion vectors.

With its complex actions and intricate movements, ActionAtlas can be a test-bed for all these ideas in video-language modeling. Moreover, reporting metrics such as the number of tokens and sampled frames can further shed light on the density of sampling from the video data and the amount of compression that is happening in tokenization.

**The videos on the web as training set.** As noted in previous work, such as the MMLU benchmark [22], modern benchmarks for foundation models assume that these models acquire the knowledge to solve various tasks by training on vast amounts of web data. Similarly, the knowledge needed to recognize actions in ActionAtlas v1.0 is readily available on video platforms like YouTube. We hypothesize that if a human were to watch all 4.5 million videos we collected, they would likely be able to recognize most of the actions in ActionAtlas v1.0. Thus, ideally, training on this massive dataset would enable the model to learn this knowledge, which we leave for future work. We will also be releasing the YouTube IDs of the 4.5 million videos we crawled for large-scale pre-training.

# 6    Limitations

**Limited to sports domain.** Sports is a real-world domain characterized by complex and subtle movements. While for many of these moves, non-expert humans can easily associate a description of an action with its corresponding movements in video and recognize it, models still lag significantly behind in capturing these nuances. Similar actions are found in other real-world domains, such as cooking, arts and crafts (e.g., knitting, sewing, etc.), dance (e.g., ballet, hip hop, salsa), and medicine. With our collection pipeline in place in Action Atlas v1.0, we plan to expand our dataset to include these domains with the help of domain-experts.

**Lack of taxonomy.** Another limitation of the current version of ActionAtlas is the absence of a taxonomy for actions. A well-defined taxonomy is helpful for ensuring that the action list in the dataset is comprehensive. In future iterations, we plan to have a collaboration between large language models and domain-experts to address this. LLMs are great at proposing a broad list of actions, including those in the long tail of the action distribution, while experts are great at refining and structuring this list within an organized taxonomy.

**Small size.** Collecting video data for domains that require expertise is an extremely challenging task. The current version of ActionAtlas is smaller compared to other multimodal benchmarks like MMMU [83]. However, with our scalable pipeline, we plan to bring in domain-experts to replace the authors in the final stage of the pipeline, as shown in Figure 2, allowing us to scale up the data collection.

# 7    Conclusion

We introduced ActionAtlas v1.0 a new VideoQA benchmark for evaluating VLMs on action recognition in real-world specialized domains. To perform well on ActionAtlas, a model must be able to understand motions that span across many frames and track the individual(s) performing the action both temporally and spatially. We collected ActionAtlas using a robust and scalable pipeline that included both automatic filtering tools and techniques, such as lexical search, LLMs, and CLIP filtering, and manual filtering via crowd-workers and the authors. Results showed that many open models perform at best close to random chance, implying that while these models excel in existing video language downstream tasks, they fall short in accurately understanding complex actions and nuanced movements in videos. Proprietary models such as GPT-4o showed better performance, improving with higher frame sampling rate, but still far from achieving high accuracy on our task.

## Acknowledgments and Disclosure of Funding

We thank the organizers of Microsoft Accelerate Foundation Models Research for providing Azure credits for OpenAI models. We also thank Google for providing Google Cloud Credits to evaluate Gemini models. Our thanks also go to Ai2 for providing data storage solutions and Amazon Mechanical Turk credits. Special thanks to Bita Fathipour for their help with verifying crowd-workers' annotations, Xiang Fan and Vivek Ramanujan for their valuable feedback on collection and evaluation, and Houng Ngo for their help with setting up the crowd-sourcing studies.

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

# A    Open Model Details

- **mPLUG-Owl** [80]: is one of the first to align both image and video modalities into large language model. This is achieved with the Qformer-based module [34] that summarizes long and dense visual information with learnable tokens, which are then combined with the text queries as input to the language model.

- **Video-LLaMA**: Unlike the above work that does not integrate audio, Video-LLaMA [85] integrates two QFormers, one for video and audio branch, and aligns the output of both visual & audio encoders with LLM's embedding space. However, as discussed in §4 we don't use any audio and text signal when evaluating the model.

- **Video-LaVIT** [24] efficiently captures the dense sequence of video by representing each video as key frames along with motion vectors. Specifically, the spatio-temporal motion encoder captures the time-varying contextual information contained in extracted motion vectors, thereby significantly enhancing LLMs' ability to comprehend the intricate actions in video. The key frame and motion tokens are then adapted to the LLMs.

- **VideoChat2** [36] adopts a progressive training approach, refining the visual encoder and Qformer for LLMS using an extensive instruction tuning dataset. Setting itself apart from previous efforts, this study enhances the performance significantly across various downstream tasks by incorporating multiple instruction tuning datasets. These datasets are compiled from both public sources and new instructions generated by ChatGPT, providing a substantial boost in performance.

- **LLaVA-Next-Video** [86] efficiently adapts LLaVA [41] to efficiently pass in long sequence of videos with high resolution with their AnyRes algorithm. supporting evidence.

- **Qwen2-VL-7B** [64] Following the official instructions for running the model, we set a limit on the total number of pixels per frame. We began with $(28 \times i) \times (28 \times i)$, where $i = 12$, to establish the maximum pixel count for each frame. If the video exceeded the model's context length after selecting frames at this resolution, we gradually decreased the value of $i$, one at a time, until the total length was compatible with the model's context capacity.

Table 7 further includes the architecture details and the input question prompt used for the open models during evaluation. We use the following system prompt for the models: "*Carefully watch the video and pay attention to the cause and sequence of events, the detail and movement of objects, and the action and pose of persons. Based on your observations, select the best option that accurately addresses the question.*" The input question and multiple choice options are formulated as "Question: {question} Choices: {choices}", and the output response is parsed to acquire the correct letter choice.

# B    Frames Per Second (FPS) Based Results

We assessed the models using FPS-based sampling, where a consistent number of frames is extracted from each second of the video. The results of this evaluation are presented in Table 5. During this process, we observed failures with numerous models when the number of frames were extremely large. Consequently, when the sampling exceeded 100 frames, we sub-sampled 100 evenly spaced frames from the sampled set.

# C    Higher Frame Detail with GPT-4o.

GPT-4 Vision API can process images (or a sequence of images) in two modes: 1. `detail: low` wherein each image is encoded into 85 tokens. 2. `detail: high` in which images are first rescaled and split into tiles where each tile is encoded into 170 tokens. [51]. Table 6 shows the results of running GPT-4 family of models with the high detail setup. The improvements are more notable when sampling only one or two frames per second.

Table 5: **Results when sampling a fixed number of frames per second from the video.**

| Model | FPS | Accuracy |
|---|---|---|
| **Qwen2-VL-7B** | 1 | 26.09 ±2.75 |
| | 2 | 28.91 ±3.00 |
| | 4 | 28.66 ±2.75 |
| **GPT-4o** | 1 | 40.00 ±2.85 |
| | 2 | 43.54 ±3.18 |
| | 4 | 42.83 ±3.11 |
| **Gemini Pro 1.5** | 1 | 32.26 ±2.92 |
| | 2 | 33.43 ±2.81 |
| | 4 | 32.20 ±3.08 |
| **GPT-4o-mini** | 1 | 30.79 ±3.06 |
| | 2 | 32.14 ±2.91 |
| | 4 | 30.16 ±2.86 |

Table 6: **Processing images with higher details using GPT4 suite of models.**

| Model | # Input frames | Low detail Acc. (%) | High detail Acc. (%) |
|---|---|---|---|
| GPT-4o | 1 | 33.08 ±2.89 | 37.96 ±3.06 |
| | 2 | 31.49 ±3.02 | 36.37 ±3.08 |
| | 4 | 39.50 ±3.10 | 39.53 ±3.13 |
| | 8 | 41.55 ±3.01 | 42.41 ±3.12 |
| | 16 | 42.95 ±2.91 | 43.90 ±3.21 |
| | 32 | 41.44 ±3.11 | 43.33 ±3.07 |
| GPT-4o-mini | 1 | 30.15 ±2.74 | 28.83 ±2.71 |
| | 2 | 27.84 ±2.90 | 29.69 ±2.92 |
| | 4 | 33.42 ±2.70 | 30.71 ±2.98 |
| | 8 | 30.20 ±2.89 | 31.19 ±3.14 |
| | 16 | 32.20 ±2.76 | 29.02 ±2.68 |
| | 32 | 31.17 ±2.90 | 29.38 ±2.94 |

Table 7: **Architecture and prompt details of open models.**

| Model | LLM | Visual Encoder | Image Size | Question Prompt |
|---|---|---|---|---|
| mPLUG-Owl [80] | LLAMA-7B [65] | CLIP ViT-L/14 [53] | 224 | Only give the best option. |
| VideoChatGPT [44] | Vicuna-7B-v1.1 [6] | CLIP ViT-L/14 [53] | 224 | Answer with the option's letter from the given choices directly. |
| VideoLLaMA [85] | LLAMA2-7B [66] | EVA ViT-G/14 [61] | 224 | Only give the best option. |
| Video-LaVIT [24] | LLAMA2-7B [66] | EVA ViT-G/14 [61] | 224 | Only give the best option. |
| VideoChat2 [36] | Vicuna-7B-v0 [6] | UMT-L [35] | 224 | *Only give the best option.* |
| LLaVA-Next-Video [86] | Vicuna-7B-v1.5 [6] | CLIP ViT-L/14-336 [53] | 336 | Answer with the option's letter from the given choices directly. |
| Qwen2-VL-7B [70] | Qwen2-7B [64] | OpenCLIP ViT-bigG [23] | 336 | Answer the given question according to the video. Only output the choice number and nothing else. |

# D   What does Gemini API Leak about the Model?

When investigating Gemini models exposed as Gemini API and the Vertex AI web application, we noticed that they might leak some information about how Gemini processes multi-modal inputs:

1. Figure 7 shows a screenshot of Google's Vertex web app. When feeding an image the token count is always 258, regardless of resolution. Therefore, if the number of tokens shown is accurate (which might not be) this could imply all images are resized to a certain size before feeding to the model. One hypothesis could be that there are $16 \times 16$ patches that are fed to the model with two indicator tokens such as "<IMAGE>" and "</IMAGE>".

2. With videos, we observed that the only factors affecting the token count were the video duration and frame rate. If a video had $N$ frames, the token count shown was always $\lfloor N/FPS \rfloor \times 265$. Therefore, according to the web app, each still image takes 258 tokens and each video frame takes 265 tokens. Those extra tokens in videos might be the timestamp tokens accompanying each frame.

3. An unusual observation was made when we uploaded a video with fewer frames than the designated FPS: the token count displayed was zero. Despite this, the model was still able to process and somewhat accurately describe the video's content. This might suggest that the web application calculates the token count offline using a predetermined formula, rather than counting the actual tokens provided to the model.

4. One possible implication of the above findings is that the video model consistently samples one frame per second when processing videos. We investigated further and were able to recover the exact frames that model samples from videos. If video's frame rate is $N$, then Gemini models select the middle frame from each second. Therefore the indices of sampled frame numbers will be $N/2, N/2 + N, N/2 + 2N , N/2 + 3N$ and so forth.

5. To verify the above claim, one approach is to insert random still images into a regular video at those frame positions. When this modified video is given to the model with a prompt such as "Exactly describe what's happening in this video without omitting any details" the model only describes the inserted still images, ignoring the rest of the video. Alternatively, the model might respond with something like "The provided video is a still image and doesn't contain any motion to describe." This pattern was consistently observed every time we tested the input in this manner.

6. Even if the original video contains inappropriate content (e.g., NSFW material), replacing frames at those positions with random images results in the model only describing the inserted images. However, should one of these frames be replaced with an inappropriate image, the model refrains from providing any output.

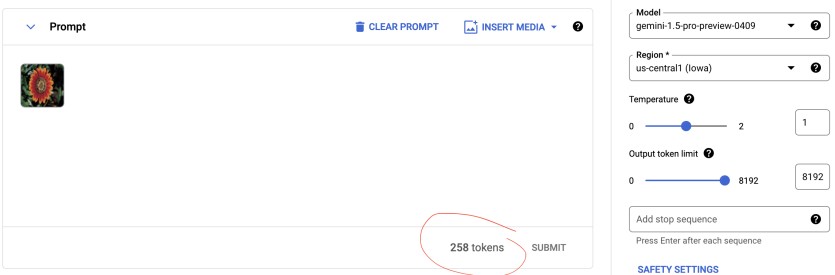

Figure 7: Screenshot of Google's Vertex AI web app.

# E   Gemini refusals

Table 8 shows Gemini Pro 1.5 refusals on ActionAtlas with different types of inputs. In general, we notice much lower refusal rate with video inputs. The refusal rate increases noticeably when the model uses chain-of-thought reasoning or sampled frames as inputs. It also increases when more number of frames are fed as the input to the model. The majority of refusals were due to dangerous content, primarily because of wrestling moves in the dataset. However, since videos of wrestling actions are readily accessible on YouTube, it's unclear why the AI model deems these videos harmful.

Table 8: **Gemini 1.5 Pro latest refusals on ActionAtlas with different setups.**

| Input | Sexually explicit | Hate speech | Harrassment | Dangerous content | Other | Total |
|---|---|---|---|---|---|---|
| Video (1 fps) | 0 | 0 | 0 | 9 | 4 | 13 (1.39%) |
| Video (all frames) | 0 | 0 | 0 | 1 | 25 | 26 (2.78%) |
| 1 frame | 4 | 0 | 0 | 44 | 0 | 48 (5.14%) |
| 2 frames | 4 | 1 | 0 | 41 | 3 | 49 (5.25%) |
| 4 frames | 4 | 0 | 0 | 46 | 4 | 54 (5.78%) |
| 8 frames | 4 | 0 | 1 | 47 | 6 | 58 (6.21%) |
| 16 frames | 4 | 0 | 0 | 50 | 14 | 68 (7.28%) |
| 32 frames | 4 | 0 | 1 | 48 | 15 | 68 (7.28%) |
| Video + Choice description | 2 | 0 | 0 | 5 | 4 | 11 (1.18%) |
| Video + CoT | 1 | 2 | 3 | 43 | 10 | 59 (6.32%) |
| Video + CoT + Choice description | 1 | 0 | 3 | 47 | 5 | 56 (5.99%) |

## F    Prompts

I'll give you a sport name and you have to generate a list of physical actions that are commonly associated with that sport.
1. Only list actions that are well-known but the list should be as exhaustive as possible.
2. If an action has multiple types list all of them. For instance in soccer there are different types of shoots such as Standard Shot (Instep Drive), Chip Shot, Curve Shot, Knuckleball Shot etc. Output all of them and each type should be in a new line.

EXAMPLE:
---
SPORT: golf
RESPONSE:
Drive/tee shot
Fairway shot
Approach shot
Chip shot
Putt
Bunker shot
Pitch shot
Flop shot
Punch shot
Recovery shot
---
SPORT: {sport}
RESPONSE:\n
"""

Figure 8: **GPT4 Prompt used for finding initial actions in different sports.**

I give you an initial list of actions in {sport}. YOU HAVE TO EXPAND THIS LIST AND GIVE A COMPREHENSIVE LIST OF ALL KNOWN ACTIONS, SHOTS, MOVES, ETC. IN THIS SPORT.
It's crucial that you include all the well known physical actions, shots, and moves specially those that might have a Wikipedia page.
Rules:
1. Give a list without description, without bullets and numbers, and just the action names line by line.
2. Optimize the list for Youtube search, so don't make the action name too long.
3. do not use parentheses, or slashes in your lines. For instance, if the action has multiple names such as "Standard Shot (Instep Drive)" then write "Standard Shot" and "Instep Drive" in two separate lines. Also do not write more description about an action in parentheses, just the action name.
4. Do not categorize the actions, just give a simple plain list of action names nothing else.
Here is my list, rewrite and expand it:
{actions}

Figure 9: **GPT4 Prompt used for expanding the action list.**

I give you a list of possible actions in {sport}. Your task is to specify which one of them are PHYSICAL actions that require MOVEMENT that can be captured in a video. Also the action has to be specific and not a general term in that sport.
Here are some examples for the kinds of actions I am looking for in a few example sports:
Alley-oop dunk in basketball
Around the world in soccer
Cross in soccer
Cruyff turn in soccer
Offensive rebound in basketball
Panenka in soccer
---
I give you 10 possible actions in {sport} and only write the name of those that are physical with movement in separate lines. Only output the exact name of actions nothing else. If none of the actions met the criteria output "".
{actions}

Figure 10: **GPT4 Prompt used for shrinking the list and removing non-physical actions.**

Write some hard negatives for move {action} in sport {domain}.
The negatives should be plausible and EXTREMELY hard to distinguish from the correct answer. However, THEY MUST BE WRONG AND DIFFERENT from the correct one. Also, the hard negatives must be well-known {domain} moves.
for each action, write 9 hard negatives. and write one hard negative in a line without any bullet points or numbers.
----
EXAMPLE:
ACTION: windshield wiper forehand
VERY HARD NEGATIVES:
Inside-out forehand
Topspin lob
Slice backhand
Flat serve
Kick serve
Reverse forehand
Volley at the net
Drop shot
Two-handed backhand
----
ACTION: {action}
VERY HARD NEGATIVES:

Figure 11: **GPT4 Prompt used for writing hard negatives for an action.**

Answer the given question according to the video. Only output the choice number and nothing else. When answering the question consider all legal and illegal moves and drills.\n{question}\n{options}

Figure 12: **Final prompt used for evaluating proprietary models.**

I will give you some information about a sport video, and you should generate a question based on the info.
The information:
1. an action.
1. description of the person performing the action.
3. what happens before the action.
4. what happens after the action.
Note that 3 and 4 could be "none". If both are "none", then just focus on the action and the person performing the action.
NOTE THAT YOU MUST NOT REFER TO THE ACTION NAME IN YOUR QUESTION!!
---
Example1:
ACTION: alley-oop dunk
PLAYER: player number 34 wearing white jersey
BEFORE: player number 34 runs towards the basket
AFTER: none
QUESTION: What best describes the move made by player wearing white jersey with number 34 after they run towards the basket?
---
Example2:
ACTION: Throwing
PLAYER: the man in the camo shirt and black pants
BEFORE: none
AFTER: none
QUESTION: What best describes the action that the man in the camo shirt and black pants performs?
---
Example3:
ACTION: Hedge
PLAYER: It is a man with a white headband and the number 34 on his jersey
BEFORE: He was guarding number 32 on the opponent team
AFTER: The opponent loses the ball
QUESTION: What best describes the action that the man with a white headband and the number 34 on his jersey performs after he was guarding number 32 on the opponent team and before the opponent loses the ball?
---
Example
ACTION: {action}
PLAYER: {player}
BEFORE: {before}
AFTER: {after}
QUESTION:

Figure 13: **GPT4 Prompt used for writing questions about the video segments.**

## G   Link to Dataset

**Google Drive**    The link to the jsonl file containing the metadata: `https://drive.google.com/file/d/1ueh5gqYgOWqQ_CFxjxsjcn8rx9wwN9Gi/view?usp=drive_link`

**HuggingFace**   `https://huggingface.co/datasets/mrsalehi/ActionAtlas-v1.0`

# H Datasheet

## H.1 Motivation

- **For what purpose was the dataset created?** The main purpose of creating ActionAtlas was to evaluate state-of-the-art VLMs on identifying domain-specialized actions. Correctly recognizing such actions necessitates the following capabilities which we believe were missing in previous video datasets, especially action recognition datasets: 1. High frame sample rate to catch fine motions in the action. 2. Correctly tracking the action actor in both time and space across the frames.

- **Who created the dataset (e.g., which team, research group) and on behalf of which entity (e.g., company, institution, organization)?** The dataset is created by RAIVN lab at the University of Washington.

- **Who funded the creation of the dataset?** The project was funded by Microsoft Accelerate Foundation Models Research program, Google, University of Washington, and Allen Institute for Artificial Intelligence.

## H.2 Composition

- **What do the instances that comprise the dataset represent (e.g., documents, photos, people, countries)?** Each instance represents a fine-grained action in some sports which consists of a video, a question, and four or five multiple choice choices from which only one is correct.

- **How many instances are there in total (of each type, if appropriate)?** There are 934 video-MCQ pairs in the dataset.

- **Does the dataset contain all possible instances or is it a sample (not necessarily random) of instances from a larger set?** No, the dataset is not a sample of a larger datset.

- **What data does each instance consist of?** Each instance consists of a video, a question, five multiple choice options, and a ground truth answer which is the option number of the ground truth action.

- **Is there a label or target associated with each instance?** Yes, the label for each instance is the correct choice for the question.

- **Is any information missing from individual instances?** No.

- **Are relationships between individual instances made explicit (e.g., users' movie ratings, social network links)?** No, the videos are sourced from different authors and creators on YouTube.

- **Are there recommended data splits (e.g., training, development/validation, testing)?** The dataset only consists of a test set.

- **Are there any errors, sources of noise, or redundancies in the dataset?** We employed extensive filtering mechanisms including automatic and AI tools and filtering by crowdworkers and authors to eliminate any potential errors and noise in the data. Some videos in the dataset might be different segments from the same original YouTube video.

- **Is the dataset self-contained, or does it link to or otherwise rely on external resources (e.g., websites, tweets, other datasets)?** The metadata is self-contained with the links to videos on YouTube.

- **Does the dataset contain data that might be considered confidential (e.g., data that is protected by legal privilege or by doctor–patient confidentiality, data that includes the content of individuals' non-public communications)?** No.

- **Does the dataset contain data that, if viewed directly, might be offensive, insulting, threatening, or might otherwise cause anxiety?** No, all the videos are segments of already available and public YouTube videos and they are already filtered by YouTube to remove harmful content.

- **Does the dataset identify any subpopulations (e.g., by age, gender)?** No.

- **Is it possible to identify individuals (i.e., one or more natural persons), either directly or indirectly (i.e., in combination with other data) from the dataset?** As the videos are sport videos sourced from YouTube, there is a possibility of recognizing famous athletes in the videos. However, when writing questions, we did not use the name of individuals in the videos; instead, we refer to them by general attributes, such as color or number of the jersey. For more details refer to §3.

- **Does the dataset contain data that might be considered sensitive in any way (e.g., data that reveals race or ethnic origins, sexual orientations, religious beliefs, political opinions or union memberships, or locations; financial or health data; biometric or genetic data; forms of government identification, such as social security numbers; criminal history)?** No.

## H.3 Collection Process

- **How was the data associated with each instance acquired?** The data was sourced from YouTube.

- **What mechanisms or procedures were used to collect the data (e.g., hardware apparatuses or sensors, manual human curation, software programs, software APIs)?** We used softwares such as Elasticsearch, GPT4, Whisper, Amazon Mechanical Turk to collect the data.

- **Who was involved in the data collection process (e.g., students, crowd-workers, contractors) and how were they compensated (e.g., how much were crowd-workers paid)?** The student authors and crows-workers. We adjusted the price per task so that the workers could make $15 per hour as the minumum wage.

- **Over what timeframe was the data collected?** The data was collected mainly between January 2024 and June 2024.

- **Were any ethical review processes conducted (e.g., by an institutional review board)?** Yes, we got IRB approval for crowd-sourcing on Amazon Mechanical Turk from University of Washington.

- **Did you collect the data from the individuals in question directly, or obtain it via third parties or other sources (e.g., websites)?** We requested crowd-workers to write questions about the given videos and we do not collect any personal data from them.

- **Has an analysis of the potential impact of the dataset and its use on data subjects (e.g., a data protection impact analysis) been conducted?** The dataset is unlikely to affect the crowd-workers. Moreover, for the individuals featured in the videos, we refrained from using any personally identifiable information (PII) like names in the questions. Instead, we referred to them using general attributes such as jersey numbers and clothing colors.

## H.4 Preprocessing/cleaning/labeling

- **Was any preprocessing/cleaning/labeling of the data done (e.g., discretization or bucketing, tokenization, part-of-speech tagging, SIFT feature extraction, removal of instances, processing of missing values)?** We did many rounds of filtering and cleaning which are discussed in Section 3 of the paper to make sure the data is of high quality. The final videos used in the dataset are raw mp4 videos.

- **Was the "raw" data saved in addition to the preprocessed/cleaned/labeled data (e.g., to support unanticipated future uses)?** The raw videos are available on YouTube as an external source.

- **Is the software that was used to preprocess/clean/label the data available?** Yes. For a thorough description of software used refer to section 3 of the paper.

## H.5 Uses

- **Has the dataset been used for any tasks already?** No.

- **Is there a repository that links to any or all papers or systems that use the dataset?** No.

- **What (other) tasks could the dataset be used for?** The dataset could be used for video tasks such as Video Understanding, Video Question Answering, and Video Compression.
- **Is there anything about the composition of the dataset or the way it was collected and preprocessed/cleaned/labeled that might impact future uses?** No.
- **Are there tasks for which the dataset should not be used?** No.

### H.6 Distribution

- **Will the dataset be distributed to third parties outside of the entity (e.g., company, institution, organization) on behalf of which the dataset was created?** No.
- **How will the dataset will be distributed (e.g., tarball on website, API, GitHub)?** On the dataset's website, Huggingface datasets, and Github.
- **When will the dataset be distributed?** We plan to release the dataset publicly by the end of October 2024.
- **Will the dataset be distributed under a copyright or other intellectual property (IP) license, and/or under applicable terms of use (ToU)?** The current version of the dataset is licensed under Creative Commons Attribution 4.0.
- **Have any third parties imposed IP-based or other restrictions on the data associated with the instances?** No.
- **Do any export controls or other regulatory restrictions apply to the dataset or to individual instances?** No.

### H.7 Maintenance

- **Who will be supporting/hosting/maintaining the dataset?** The dataset will be hosted on our website, GitHub repository, Huggingface, and Google drive.
- **How can the owner/curator/manager of the dataset be contacted (e.g., email address)?** Email address.
- **Is there an erratum?** No.
- **Will the dataset be updated (e.g., to correct labeling errors, add new instances, delete instances)?** Yes, we plan to update the data for any potential errors that will be discovered in the future.
- **If the dataset relates to people, are there applicable limits on the retention of the data associated with the instances (e.g., were the individuals in question told that their data would be retained for a fixed period of time and then deleted)?** No.
- **Will older versions of the dataset continue to be supported/hosted/maintained?** Most likely yes.
- **If others want to extend/augment/build on/contribute to the dataset, is there a mechanism for them to do so?** Yes, we plan to implement such mechanisms on the website of our dataset.

## I  License

The current version of the dataset is licensed under Creative Commons Attribution 4.0.

## J  Author Statement

The authors bear all responsibility in case of violation of rights and confirmation of the data license.

## K  Mechanical Turk HITs

Figures 14 and 15 shows the templates and the instructions used for verification and localization of actions with the help of crowd-workers on Amazon Mechanical Turk. For both tasks we calibrated the price per hit so that the workers could earn $15 per hour which is the minimum wage.

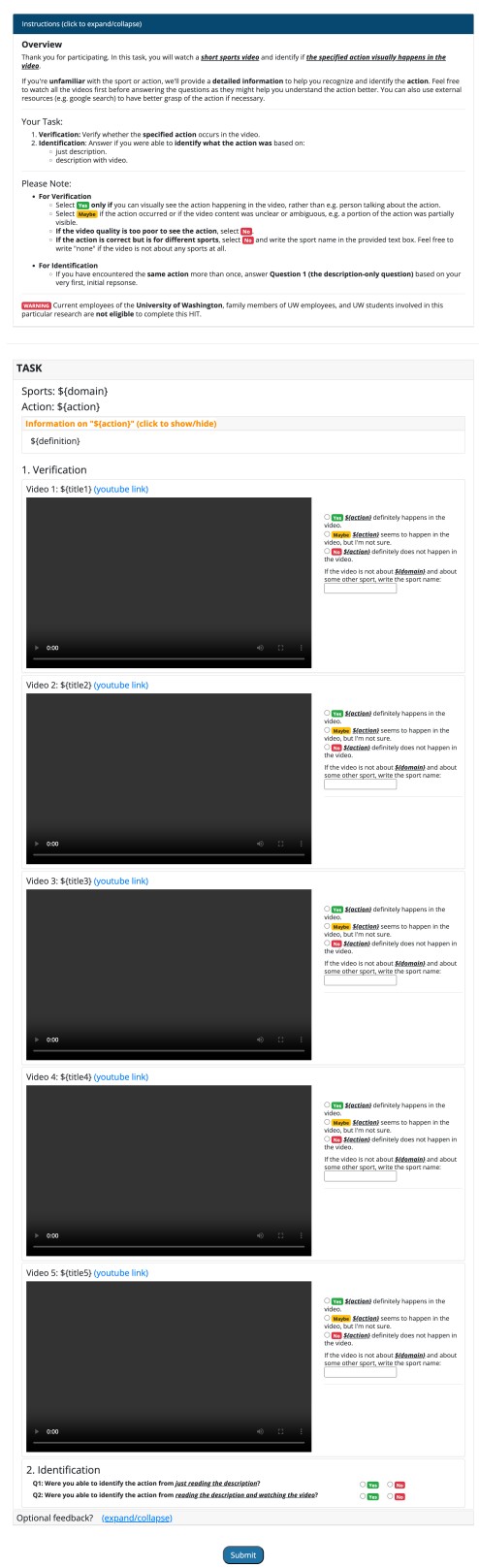

Figure 14: **Template used for Verifying presence of actions by crowd-workers.**

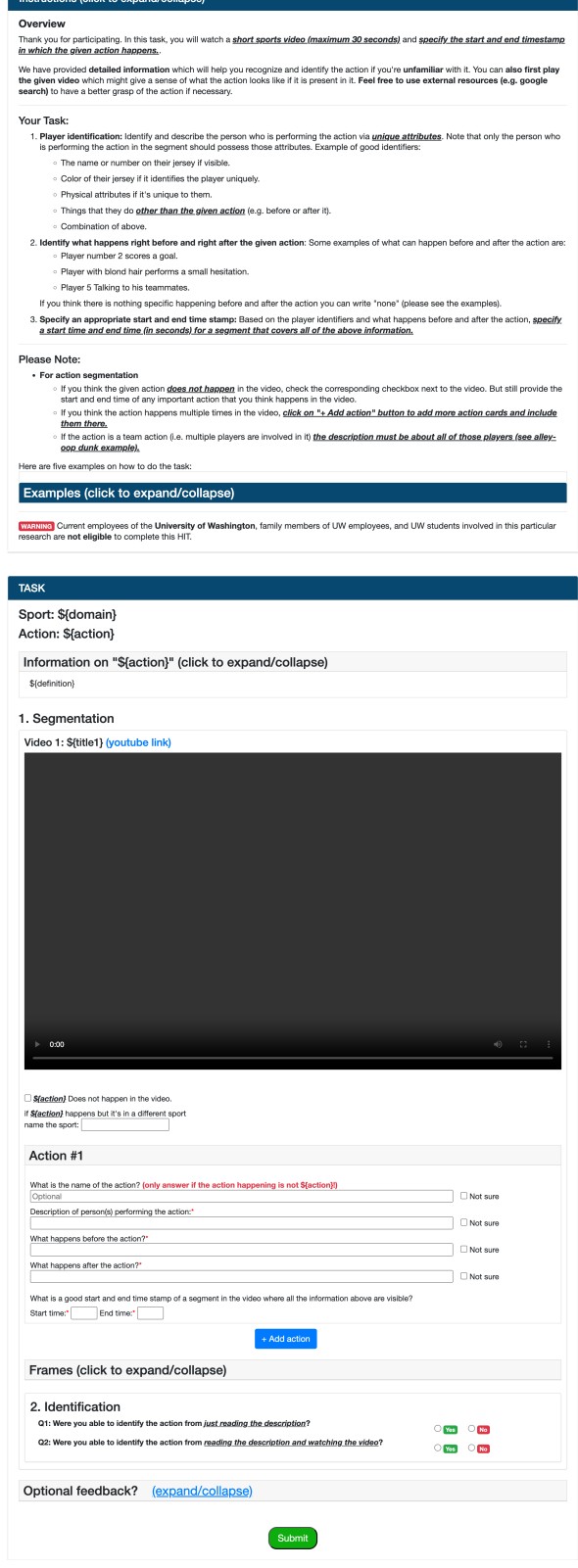

Figure 15: **Template used for localizing actions in 30 second segments.**

