# A  Open Model Details

- mPLUG-Owl [58] is one of the first to align both image and video modalities to large language model. This is achieved with the Qformer-based abstractor module [23] that summarizes long and dense visual information with learnable tokens, which are then combined text queries as input to the language model.

- VideoChatGPT adapts LLaVA [29], an image-base visual instruction tuned model, to video understanding tasks by temporally pooling the sequence of frame embeddings to get the video-level features. These features are projected by linear layer as language embedding tokens and passed down to language model. The model is trained with 100,000 video-instruction pairs annotated by language models.

- Unlike the above work that does not integerate audio, VideoLLAMA [60] integrates two QFormers, one for video and audio branch, and aligns the output of both visual & audio encoders with LLM's embedding space.

- VideoLaVIT [17] efficiently captures the dense sequence of video by representing each video as keyframes and temporal motions. Specifically, the spatiotemporal motion encoder captures the time-varying contextual information contained in extracted motion vectors, thereby significantly enhancing LLMs' ability to comprehend the intricate actions in video. The key frame and motion tokens are then adapted to the LLMs.

- VideoChat2 [25] progressively trains the visual encoder and Qformer to LLMS, with the comprehensive instruction tuning dataset. Unlike prior work, the work adds multiple set of instruction tuning dataset curated from public dataset and newly instructions generated by ChatGPT, leading to huge boost in performance across diverse downstream task.

- LLaVA-Next-Video [62] efficiently adapts LLaVA [29] to efficiently pass in long sequence of videos with high resolution with their AnyRes algorithm. It also introduces DPO [38, 61] variant of the model trained by the preference data generated by LLM, where videos are represented with their detailed captions as supporting evidence.

Table 2: Architecture details of open source models and question prompts used in the input text.

| Model | LLM | Visual Encoder | Image Size | Question Prompt |
|---|---|---|---|---|
| mPLUG-Owl [58] | LLAMA-7B [48] | CLIP ViT-L/14 [36] | 224 | Only give the best option. |
| VideoChatGPT [31] | Vicuna-7B-v1.1 [4] | CLIP ViT-L/14 [36] | 224 | *Answer with the option's letter from the given choices directly.* |
| VideoLLaMA [60] | LLAMA2-7B [49] | EVA ViT-G/14 [45] | 224 | Only give the best option. |
| Video-LaVIT [17] | LLAMA2-7B [49] | EVA ViT-G/14 [45] | 224 | Only give the best option. |
| VideoChat2 [25] | Vicuna-7B-v0 [4] | UMT-L [24] | 224 | *Only give the best option.* |
| LLaVA-Next-Video [62] | Vicuna-7B-v1.5 [4] | CLIP ViT-L/14-336 [36] | 336 | *Answer with the option's letter from the given choices directly.* |

Table 2 further includes the architecture details and the input question prompt used for the open models during evaluation. We use the following system prompt for all the models: "*Carefully watch the video and pay attention to the cause and sequence of events, the detail and movement of objects, and the action and pose of persons. Based on your observations, select the best option that accurately addresses the question.*" The input question and multiple choice options are formulated as "Question: {question} Options: {choices}", and the output response is parsed to acquire the correct letter choice.

 **B   Prompts**

I'll give you a sport name and you have to generate a list of physical actions that are commonly associated with that sport.
1. Only list actions that are well-known but the list should be as exhaustive as possible.
2. If an action has multiple types list all of them. For instance in soccer there are different types of shoots such as Standard Shot (Instep Drive), Chip Shot, Curve Shot, Knuckleball Shot etc. Output all of them and each type should be in a new line.

EXAMPLE:
---
SPORT: golf
RESPONSE:
Drive/tee shot
Fairway shot
Approach shot
Chip shot
Putt
Bunker shot
Pitch shot
Flop shot
Punch shot
Recovery shot
---
SPORT: {sport}
RESPONSE:\n
"""

Figure 6: GPT4 Prompt used for finding initial actions in different sports.

I give you an initial list of actions in {sport}. YOU HAVE TO EXPAND THIS LIST AND GIVE A COMPREHENSIVE LIST OF ALL KNOWN ACTIONS, SHOTS, MOVES, ETC. IN THIS SPORT.
It's crucial that you include all the well known physical actions, shots, and moves specially those that might have a Wikipedia page.
Rules:
1. Give a list without description, without bullets and numbers, and just the action names line by line.
2. Optimize the list for Youtube search, so don't make the action name too long.
3. do not use parentheses, or slashes in your lines. For instance, if the action has multiple names such as "Standard Shot (Instep Drive)" then write "Standard Shot" and "Instep Drive" in two separate lines. Also do not write more description about an action in parentheses, just the action name.
4. Do not categorize the actions, just give a simple plain list of action names nothing else.
Here is my list, rewrite and expand it:
{actions}

Figure 7: GPT4 Prompt used expanding the action list.

I give you a list of possible actions in {sport}. Your task is to specify which one of them are PHYSICAL actions that require MOVEMENT that can be captured in a video. Also the action has to be specific and not a general term in that sport.
Here are some examples for the kinds of actions I am looking for in a few example sports:
Alley-oop dunk in basketball
Around the world in soccer
Cross in soccer
Cruyff turn in soccer
Offensive rebound in basketball
Panenka in soccer
---
I give you 10 possible actions in {sport} and only write the name of those that are physical with movement in separate lines. Only output the exact name of actions nothing else. If none of the actions met the criteria output "".
{actions}

Figure 8: GPT4 Prompt used for shrinking the list and removing non-physical actions.

## C  Jail-breaking Multi-modal Gemini

When investigating Gemini models on the Vertex AI web app, we noticed that it might leak some information about how Gemini processes multi-modal inputs:

1. Figure 12 shows a screenshot of Google's Vertex web app. When feeding an image the token count is always 258, regardless of resolution. Therefore, if the number of tokens shown is accurate (which might not be) this could imply all images are resized to a certain size before feeding to the model. One hypothesis could be that there are $16 \times 16$ patches that are fed to the model with two indicator tokens such as "<IMAGE>" and "</IMAGE>".

2. With videos, we noticed that the only factor that seemed to matter in token count was the video length in time. If a video had $N$ frames, the token count shown was always $\lfloor N/FPS \rfloor \times 265$. Therefore, according to the web app, each still image takes 258 tokens and each video frame takes 265 tokens. Those extra tokens in videos might be the timestamp tokens accompanying each frame.

3. Another unusual observation was that when we uploaded a video with fewer frames than the FPS, the token count shown is zero. Yet, the model still processes and describes what's in the video somewhat correctly. This could potentially indicate that the web app calculates number of tokens offline using a predetermined formula without counting the actual tokens that are fed to the model.

4. One potential implication of the above observations is that the video model always sample one frame per second when processing videos. We investigated further and were able to recover the exact frames that model samples from videos. If the frame rate of the video is $N$, then Gemini samples middle frame from each second. Therefore the indices of sampled frame numbers will be $N/2$, $N/2 + N$, $N/2 + 2N$ , $N/2 + 3N$ and so forth.

5. The way to test the above claim is to inject some random images inside a regular video at those positions. When you feed such inputs to the model and ask the model to describe it with a prompt such as "Exactly describe what's happening in this video. Don't leave out any details" the model only describes still images and nothing about the video; or outputs a response such as ""The provided video is a still image and does not contain any video or movement to describe". We could reproduce this behavior every time we fed the input.

6. Even if the video is an unsafe content (e.g. NSFW), by changing those specific frames, the model describes only those injected images. However, if one the frames at those positions is changed to an unsafe image the model does not output anything.

Figure 9: GPT4 Prompt used for writing questions about the video segments.

# D   Mechanical Turk HITs

Figures 14 and 15 shows the templates and the instructions used for verification and localization of actions with the help of crowd-workers on Amazon Mehcanical Turk. For both tasks we calibrated the price per hit so that the workers could earn $15 per hour which is the minimum wage.

Write some hard negatives for move {action} in sport {domain}.
The negatives should be plausible and EXTREMELY hard to distinguish from the correct answer. However, THEY MUST BE WRONG AND DIFFERENT from the correct one. Also, the hard negatives must be well-known {domain} moves.
for each action, write 9 hard negatives. and write one hard negative in a line without any bullet points or numbers.
----
EXAMPLE:
ACTION: windshield wiper forehand
VERY HARD NEGATIVES:
Inside-out forehand
Topspin lob
Slice backhand
Flat serve
Kick serve
Reverse forehand
Volley at the net
Drop shot
Two-handed backhand
----
ACTION: {action}
VERY HARD NEGATIVES:

Figure 10: GPT4 Prompt used for writing hard negatives for an action.

Answer the given question according to the video. Only output the choice number and nothing else. When answering the question consider all legal and illegal moves and drills.\n{question}\n{options}

Figure 11: Final prompt used to evaluate proprietary models.

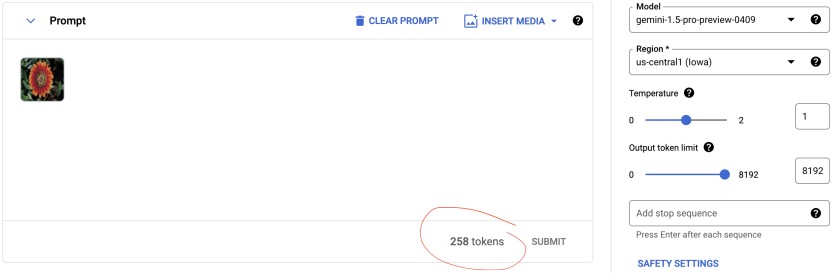

Figure 12: Screenshot of Google's Vertex AI web app.

## E    Link to Dataset

**Google Drive**    The link to the main dataset jsonl file: https://drive.google.com/file/d/1TtJ2hu6etf8js7RzWaRGBWiTSKDLSTRP/view?usp=sharing
The metadata file containing information about the keys of each object in the jsonl file: https://drive.google.com/file/d/1zONJO-Xdhp9A23U-7gm9vZaXNZQs3p4R/view?usp=sharing

**Note**: The data is in JSONL format, which is a widely recognized format. The metadata corresponding to our dataset is simple and only contains description of the keys of objects. Because of simplicity of the data, we chose not to use tools such as ML Croissant to create the data and metadata files. We will host the data on Huggingface and our GitHub repository.

## F  Dataset Statistics

**Number of actions:** 284
**Number of videos:** 557
**Number of sports:** 43
**Average length of videos:** 5.55 seconds
**Average frame rate of videos:** 32.7 FPS
**Distribution of actions per sport**:

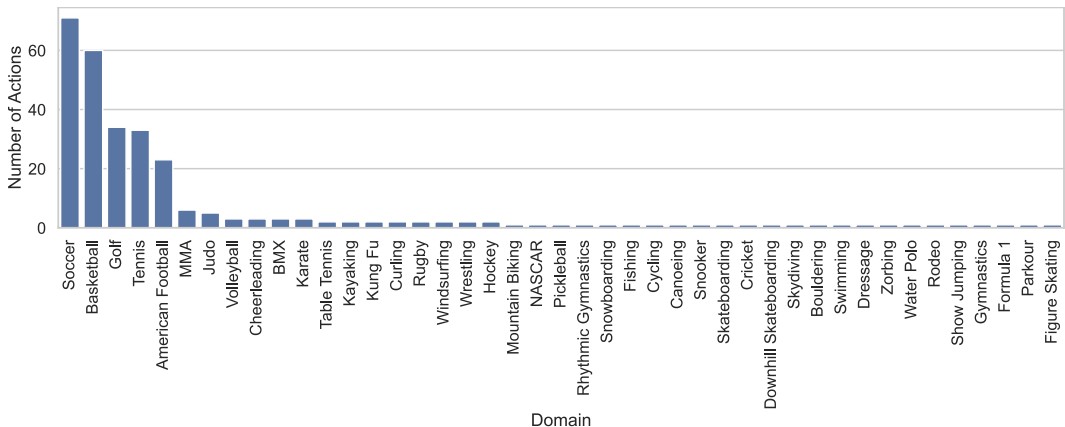

Figure 13: **Distribution of actions across sports.**

## G  Datasheet