# OpenReview forum: "ActionAtlas: A VideoQA Benchmark for Domain-specialized Action Recognition"
_NeurIPS.cc/2024/Datasets_and_Benchmarks_Track — NeurIPS 2024 Track Datasets and Benchmarks Poster_

### Official Review · Reviewer_DiB7 · 2024-07-05
**Nice work, but concerns in the evaluation**

**Rating:** 8
**Confidence:** 4
**Clarity:** Yes, the writing is clear and all the…

**Review:**

Overall, I think the work is novel because the new dataset provides a very challenging benchmark that requires discerning subtle action, understanding specific techniques/skills, and holistic video understanding. The motivation and necessity of the benchmark are clear, and the actual dataset also reflects the aspects above. While the failure of open models is rather unsurprising because of its requirements for expert knowledge, providing one specific example that the current best models struggle with is valuable to promote further research on more advanced video-centric models. Specializing in sports is potentially a drawback but it won’t lose the essence of the underlying challenges.

On the other hand, there are some weaknesses out here:
* (i) Mixed challenges in language/video understanding: One major weakness is that the proposed benchmark mixes two problems (a) understanding the meaning of technical skill names in each sport and (b) understanding the actual visual content after understanding technical skills. If the question is given as in Figure 1, without any explanation of each skill name, it is natural for the open model to fail simply because they are primarily trained by existing video-text corpus which may not include such specialized knowledge. Similarly, the better performance of proprietary models is probably mainly due to their superior ability in language reasoning, not just visual understanding.
* (ii) Too special task: I doubt the proposed benchmark is too specific/special to discern trivial differences in special domains rather than reflecting the essential video understanding in real-world scenarios (e.g., social interaction, cooking). The work could be better if some evidence of models good at ActionAtlas is also beneficial in other video understanding tasks.
* (iii) Lack of supervised baseline: In my understanding, all the models are not fine-tuned for this specific task. Regarding the specialty of the dataset, I wonder if supervised baseline models could be employed at least in a smaller subset of the dataset.
* (iv) Unclear process of human verification: The proposed semi-automatic annotation procedure looks generally good. However, I wonder if crowd workers verifying the presence of actions by following the automatically generated definition by GPT-4 may cause errors in annotation because there is no guarantee that the GPT4-generated definition surely covers the essential points of recognizing the actions.
* (v) Fewer insights on the results: Related to (i), since the language/video understanding is mixed in the ActionAtlas benchmark, the analysis on open models was almost meaningless despite the variations in the architecture. Analysis on the number of input frames is interesting but it is quite not sure in which cases higher frame rates are required because there are almost no qualitative examples included, and only the numbers are shown in detail.

To summarize, I think the overall direction is promising and will put a clear acceptance if we can separate the mixed problem of language/video understanding, but in its current form, I think the essential challenge of visual content understanding is hidden by understanding the linguistic meaning of each special action.

**Strengths:**

* Providing a very challenging benchmark even the most powerful video (image) language models struggle is surely a contribution to the community. The dataset is carefully designed to avoid trivial shortcuts.
* The semi-automatic data collection and annotation process is a good showcase for further creating video understanding benchmarks, where it is often time-consuming to find the temporal segment of interest.
* The analysis mainly focused on the number of input frames is one of the important directions to pursue. Efficiency metrics may be also important regarding the longer context requirement in recent video understanding tasks.

**Additional Feedback:**

L039: Missing space “ActionAtlasfor”
L210: Proprietray->Proprietary
G.6: Isn’t the first question’s answer “yes” since it’ll be publicly available?

Post rebuttal: I've read through all the reviewer's messages and the author's rebuttal. As made in the response to the authors, I think the authors have cleared the critical concerns on solving mixed problems on videos and language. The dataset is unique and its quality is high enough to be a valuable resource for other researchers in this field. Therefore, I'll raise my score. Although I think the discussions on the training should be removed since it's not in the original part of the submission, releasing such data later will be helpful for the community.

**Correctness:**

Overall the dataset and benchmark construction process is clearly written and correct. The evaluation scheme is conducted in a standard manner.

**Documentation:**

The checklist and the datasheet include sufficient information on the collection process.

**Ethics:**

Despite the data is collected from publicly available YouTube videos, it seems like some videos include edits of broadcasted videos, which may not comply with the data-holder's permission (e.g., https://storage.googleapis.com/video_llm/data/action-benchmark/final_segments_no_audio_blurred/CCs5D-c1ce4_0_15_0.0_7.0.mp4). Regarding the nature of the sport domain (which may likely to include such broadcast videos), we suggest the authors to identify such videos and remove them if problematic.

**Limitations:**

Yes, the authors have mentioned their limitations.

**Opportunities For Improvement:**

* What will happen if we provide the GPT4-generated definition of each action in the question part? It may mitigate the issues of understanding the language description of each action and put more focus on recognizing the visual content.
While three example questions are provided, it is still unclear what kind of ability is required to solve this benchmark. It should include more examples including qualitative results, and also the ideal inference process for answering these questions.
* Similarly, the analysis is too biased on quantitative analysis, and I feel like it’s just showing numbers without concrete insights or potential direction toward improvement. I suggest having more qualitative analysis and discussing the actual challenge included in the ActionAtlas benchmark. Either, it would be beneficial to have detailed analysis on which phase the model is getting wrong (e.g., linguistic understanding, identifying target person, etc.).
* Sec. A: Showing the training data of each model would help understand rather than model architecture.
* Although minor, the number of actions are highly biased towards major sports (soccer, basketball, golf, tennis, and American football). For other sports the number of actions is pretty small, which may be a limitation as a general benchmark.
The average length of videos is quite short (5.55 sec), which may not capture about understanding longer temporal context. It’ll be better if the benchmark could include longer samples.

**Relation To Prior Work:**

Yes, the authors include appropriate references to prior work.

**Summary And Contributions:**

This paper introduces a new video dataset for fine-grained action understanding. The dataset is carefully designed to require a comprehensive understanding of detailed actions by looking at video frames at a higher sampling rate rather than solvable just by looking at a few frames. To construct such a challenging benchmark, a new pipeline of using LLMs to (i) find target actions (ii) clip selection and (iii) multi-choice QA generation is introduced. Experimental results using open and proprietary models including the powerful GPT-4o and Gemini reveal that the current models struggle to correctly answer the questions even with higher frame rates.

This work’s contribution lies in (i) the introduction of the ActionAtlas dataset based on Internet video curation and QA creation (ii) an efficient dataset creation pipeline using GPT-4 (iii) Extensive analysis of the performance in the ActionAtlas dataset, including some tips on evaluating closed models.

---

> ### Author Rebuttal · Authors · 2024-08-17
>
> We thank the reviewer for their detailed review. We also appreciate the reviewer's recognition of the novelty in our work, particularly the semi-automatic collection pipeline and the use of GPT4-text to identify candidate temporal segments faster. We are also pleased that they recognized the importance of analysis on the number of input frames.
>
> We address the questions and weaknesses in what follows:
>
> 1. **Mixed challenges in language/video understanding**: To better disentangle the two elements highlighted by the reviewer, as discussed in the general rebuttal, we did the following evaluation: for each question option, we generated a 2-3 sentence description using GPT-4o and included these descriptions alongside the options. When evaluating GPT4-o with 16 frames on the benchmark, the accuracy improved from 43.04% to 44.00%. The following table shows the results of the same experiment for open models with 16 frames as input:
> | Model                     | Acc. Without description | Acc. With description |
> |---------------------------|---------------------|------------------|
> | mPlug-owl                 | $19.49$%            | $20.87$%         |
> | VideoChatGPT              | $19.17$%            | $18.09$%         |
> | VideoLLAMa                | $20.13$%            | $21.84$%         |
> | Video Chat2               | $20.88$%            | $21.95$%         |
> | LLaVA-Next-video          | $20.77$%            | $21.73$%         |
> | LLaVA-Next-video-dpo      | $20.34$%            | $21.41$%         |
>
> In all models except VideoChatGPT providing descriptions leads to a small performance improvement. Note that we provided the same descriptions to non-expert workers on Mechanical Turk for human evaluation which resulted in an accuracy of 61.02%. This could indicate that the low accuracy of models is primarily due to poor visual recognition rather than a misunderstanding of the meaning of technical skill names. Further supporting this argument is the qualitative analysis presented in the rebuttal PDF.
>
> 2. **Too special task**: We believe that understanding fine-grained actions across various domains is an important aspect of real-world video understanding which has not been addressed well in the literature. To showcase this, in the first version of the benchmark, we focused on sports a domain characterized by actions involving intricate motions. We have targeted fine-grained actions in more domains, including cooking (as noted by the reviewer), arts and crafts (e.g., knitting, sewing, etc.), dancing (e.g., ballet, hip hop, salsa), medical actions (e.g, basic medical assessments). With our collection pipeline in place in Action Atlas v1.0, we plan to expand our dataset to include these domains. While existing datasets like AVA [1] and ActivityNet [2] have been created for actions in scenarios such as social interaction (e.g. sitting, talking, shaking hands), current open and proprietary models have already saturated these benchmarks. However, our observation has been that these models struggle significantly when the actions in domains are highly similar and the motions are more intricate and Action Atlas highlights that.
>
>
> 3. **Unclear process of human verification**:  To mitigate such errors, as discussed in the paper, we implemented several safeguards in our pipeline:
> **i**. When verifying the presence of actions, five 30 second candidate segments are shown to the crowd-worker. This helped the annotator to compare the videos with each other and the description given by GPT4-text. In case of any discrepancies or incomplete information, the worker is instructed to search for additional videos of the action on YouTube and other platforms. Only when the annotator is confident should they vote for the presence of the action
> **ii**. Each set of five videos for an action is reviewed by three workers, and we only used videos where workers reached a consensus on the presence of the action.
> **iii**. The authors manually reviewed all examples in the dataset to ensure that the action definition matched the action depicted in the video. For edge cases, we searched for the action on YouTube to confirm its accuracy. In rare instances, we found that the description of the action did not match the actual action (e.g., the "double pump" action in kayaking). In such cases we searched for more youtube videos and further educated ourselves and only then concluded if the action is present in the video or not.
>
> [1] Gu, C., Sun, C., Ross, D. A., Vondrick, C., Pantofaru, C., Li, Y., Vijayanarasimhan, S., Toderici, G., Ricco, S., Sukthankar, R., Schmid, C., & Malik, J. (2018). AVA: A video dataset of spatio-temporally localized atomic visual actions. arXiv. https://arxiv.org/abs/1705.08421
>
> [2] Heilbron, F. C., Escorcia, V., Ghanem, B., & Niebles, J. C. (2015). ActivityNet: A large-scale video benchmark for human activity understanding. In 2015 IEEE Conference on Computer Vision and Pattern Recognition (CVPR) (pp. 961-970). IEEE. https://doi.org/10.1109/CVPR.2015.7298698

---

> > ### Author Rebuttal · Authors · 2024-08-17
> >
> > 4. **Lack of supervised baseline**: Similar to prior modern benchmarks for large foundation models such as MMLU [3] and GPQA [4], we argue that to recognize fine-grained actions in different domains, models should be pretrained on internet scale data. Therefore, in contrast to previous traditional video benchmarks such as YouCook2, we did not curate a training set for supervised training. However, as detailed in Section 3.2 of the paper, we have compiled over 8 million videos related to these fine-grained actions, which we plan to use for model training in future work. Additionally, we will be releasing the corresponding YouTube IDs as a pretraining set for others to use.
> >
> > 5. **Fewer insights on the results**: Please see our response to the first weakness language/video understanding being mixed in the evaluation. We have included qualitative examples in the PDF and will add them to the paper.
> >
> > 6. **the analysis is too biased on quantitative analysis**: We prompt different models to provide their reasoning and why they choose an option. Please see the rebuttal PDF for such qualitative examples. Our observation shows that in most failure cases the model’s visual recognition capability is the main reason for failure and what it describes does not actually happen in the video.
> >
> > 7. **Showing the training data of each model would help understand rather than model architecture**: We will revise the paper and write about the training data of each model in the baseline section.
> >
> > 8. **The average length of videos is quite short**: While we agree with the reviewer that understanding longer temporal context is an important problem, our current focus is on recognizing fine-grained actions and the intricate moves and motions within them, as well as their connection to frame sampling rates. Although in our collection pipeline, we first collect 30 second candidate segments, to better align them with our goal, we further localize the action of interest in the video. We plan to use these 30 second segments (or potentially segments longer than 30 seconds) and have experts narrate fine-grained actions happening in them with the order of happening. Having multiple instances of fine-grained actions would make the benchmark more challenging and interesting which we will leave for future work.
> >
> > 9. **Ethics concern**: As mentioned by the ethics reviewer, we will provide metadata of these videos when releasing the dataset.
> >
> > [3] Hendrycks, D., Burns, C., Basart, S., Zou, A., Mazeika, M., Song, D., & Steinhardt, J. (2021). Measuring massive multitask language understanding. arXiv. https://arxiv.org/abs/2009.03300
> >
> > [4] Rein, D., Hou, B. L., Stickland, A. C., Petty, J., Pang, R. Y., Dirani, J., Michael, J., & Bowman, S. R. (2023). GPQA: A graduate-level Google-proof Q&A benchmark. arXiv. https://arxiv.org/abs/2311.12022

---

> > > ### Comment · Reviewer_DiB7 · 2024-08-19
> > >
> > > The reviewer thanks the authors for their detailed response.
> > >
> > > * The additional experiments on adding action description was very helpful to understand the performance of the open models. It seems like more or less the open models have significant challenges in understanding the visual contents. However, after looking at more qualitative examples, I also thought the issue may be a matter on localizing the target players in the video (e.g., "player #7" requires understanding the person with a uniform of "7" printed). Also, many failure cases may be simply caused by sparse sampling of frames. Therefore, there still exists some confusion in the model (as pointed out by the other reviewers) but I think the most critical concern has been cleared.
> > > * The plan of using the collected Internet videos for training will be a great addition. However, I'll omit it from the evaluation since it is not included in the submission.
> > > * The careful human verification process seems enough for maintaining the quality although it's not perfect because of its challenging situations.
> > > * The rebuttal PDF helped understanding the failure cases. It looks like the frame number may be the key factor in such cases, so having a discussion on that may be helpful.
> > >
> > > Overall, I'm convinced with the additional experimental results provided by the authors. Despite there are some concerns on mixed challenges included in the benchmark, I think it will be a good benchmark for studying fine-grained video action recognition task, replacing some of the existing datasets. I welcome any additional comments from the other reviewers.

---

> > > > ### Author Response · Authors · 2024-08-22
> > > > **Appreciate the positive review and feedback**
> > > >
> > > > We appreciate the reviewer's positive feedback on our new experiments and their insightful comments.

---

### Official Review · Reviewer_jE6k · 2024-07-24
**Poor Dataset Design**

**Rating:** 3
**Confidence:** 5

**Review:**

Concerns with the Paper:

1. Clarity of Objectives:
- The paper could benefit from clearer objectives. At times, it shifts focus between realigning video benchmarking, fine-grained action recognition, and evaluating temporal understanding, making it difficult to discern the primary goal.

2. Definition of Fine-Grained Actions:
- The paper does not clearly define what it means by fine-grained actions. In conventional video understanding, fine-grained action recognition involves discerning between very similar actions with subtle differences. Providing a more explicit definition could enhance understanding.

3. Objective and Dataset:
- The paper aims to study the effectiveness of recently popularized multimodal models in recognizing "fine-grained actions" and proposes a small dataset for this purpose. While the paper intends to unravel the complexity of fine-grained action recognition, some of this complexity may arise from unintended, entangled factors.

4. Poor Dataset DEsign and Unintended Factors:
- Several unintended factors seem to contribute to the overall challenge of the dataset, which might not align with the fine-grained nature of the actions as defined in the paper. These include:
- Temporal localization (Fig. 1, top).
- Unusual or weird action names.
- Confusing action names that can have different meanings depending on the context. For example, a cartwheel in kayaking differs significantly from a cartwheel in gymnastics. Given that MLLMs are more likely to encounter the gymnastics context, it is understandable why an MLLM might be unsure when answering the second question in Fig. 1. I mean what is next, swapping the labels of kayaking and soccer in the test set and see pretrained models confuse and fail?!

5. Dataset Challenges:
- The challenges presented by the dataset may not solely arise from the fine-grained nature of the actions. A combination of unintended factors may be contributing to the lower performance of multimodal language models (MLLMs).

6. Action Name Descriptions:
- It might be helpful to explore whether using more conventional action name descriptions instead of unusual action names affects model performance. If models still perform poorly, this could indicate more credibly that MLLMs struggle with fine-grained action recognition.

7. Contradictory Observations:
- Models like VideoChat2 have been trained on over 20 challenging video understanding tasks, including fine-grained action recognition, and have performed adequately. Clarifying how these observations align with the paper's findings would be beneficial.

8. Human Baseline as Gold Standard:
- Including a human baseline as a gold standard in lines 233-238 could provide valuable insights into whether additional frames enhance the credibility of the speculation.

9. Computational Complexity/Granularity:
- It would be helpful to justify the assertion in lines 236-238 that the fine-grained aspect of the actions or dataset demands higher computational complexity or granularity. A human baseline might assist in this justification.

10. Conclusion on Fine-Grained Action Recognition:
- Given the factors mentioned, it is challenging to conclude satisfactorily that the dataset is specifically challenging models in fine-grained action recognition.

11. Line 89:
- Claiming 1161 actions is a bit of a stretch. By that logic, a QA dataset with 50,000 samples and 4 options each would be like distinguishing between 200,000 actions! Come on now! That's like saying each brick counts as an entire castle.

Due to the aforementioned shortcomings, I am inclined to recommend not accepting this paper in its current form.

**Strengths:**

The problem is important, but the dataset design is poor and cannot effectively support the central research question.

**Additional Feedback:**

Please refer to Review section.

**Clarity:**

Some of the key definition/information is missing as mentioned in Review section.

**Correctness:**

No, in my assessment the dataset design is poor and cannot effectively support the central research question. For further details, please refer to Review section.

**Documentation:**

Yes,

**Ethics:**

No, but older version of the checklist is used.

**Limitations:**

Yes.

**Opportunities For Improvement:**

Included in Review section.

**Relation To Prior Work:**

Yes.

**Summary And Contributions:**

This paper aims to study the effectiveness of recently popularized multimodal models (MLLMs) at recognizing so-called "fine-grained actions." To this end, the paper proposes a small dataset. While it presents its work as unraveling the complexity of fine-grained action recognition, the complexity might largely arise from a bouquet of unintended, entangled factors. Therefore, due to the poor dataset design, we cannot effectively evaluate the ability of MLLMs to perform fine-grained action recognition.

---

> ### Author Rebuttal · Authors · 2024-08-17
>
> We address the questions and weaknesses in what follows:
>
> 1. **Clarity of Objectives:** The goal of this work is to evaluate foundation models on fine-grained action recognition within specialized domains that have practical real-world applications. To showcase this, in the first version of the benchmark, we focused on sports, a domain characterized by actions involving fine-grained intricate motions. By "temporal understanding" in the submission, we refer to the need for multiple frames to accurately recognize these actions, as opposed to relying on a single frame. This is supported by the observed improvement in model performance as the number of frames increases.
> Furthermore, as discussed in section 3.6 of the submission, certain actions can only be meaningfully recognized when temporal context is provided. For instance, in soccer, an action can be identified as a dummy when one player passes the ball, and another player lets it pass through to a teammate behind them; this scenario inherently requires temporal context to be captured in the video. However, we made sure not to confuse this with the challenge of long-form temporal understanding as we asked crowd-workers to localize the actions (stage 5 in the data collection pipeline) within 30-second segments that we found in stage 4 of our pipeline.
> We will revise the paper to reflect these objectives more clearly.
>
>
> 2. **Definition of Fine-Grained Actions:** We adhere to the conventional approach as outlined by the reviewer. We prompt GPT4 and leverage its vast knowledge about different domains to identify such actions. In the current version of the benchmark, the videos were collected in batches.  For each batch, we set a threshold for the frequency of occurrence of action names in YouTube titles, discarding those that appeared less frequently in later stages of the pipeline. This process resulted in a total of 580 actions in 56 domains.
>
> 3. **Poor Dataset Design and Unintended Factors (temporal localization Fig. 1 top)**: as discussed in section 3.6 of the submission, certain actions can only be meaningfully recognized when temporal context is provided. For instance, in soccer, an action can be identified as a dummy when one player passes the ball, and another player lets it pass through to a teammate behind them; this scenario inherently requires temporal context to be captured in the video. However, we made sure not to conflate this with the challenge of long-form temporal understanding as we asked crowd-workers to further localize the actions in shorter segments (stage 5 in the data collection pipeline) within 30-second segments that we found first.
>
> 4. **Unusual or weird action names**: We were wondering if the reviewer could clarify what they mean by “unusual or weird action names”. The action names provided in this work are established in their respective community of that domain and there are uploaded videos about these actions on youtube.
>
>
> 5. **Confusing action names that can have different meanings depending on the context**: It’s important to note that the video is provided as context and each video only depicts a single sport, not multiple sports. If an MLLM is given a gymnastics video and fails to recognize that the action name “cartwheel” in the options refers to a gymnastics move, we consider this a shortcoming of the model, not the benchmark.
>
> 6. **Action Name Descriptions**: To factor out the element of models not knowing the action names, we did an extra evaluation where the description of action is provided along with its name. When evaluating GPT4-o with $16$ frames the accuracy increased from 43.04% to 44.00%. The following table shows the results of the same experiments for open models with $16$ frames as input:
>
> | Model                     | Acc. Without description | Acc. With description |
> |---------------------------|---------------------|------------------|
> | mPlug-owl                 | $19.49$%            | $20.87$%         |
> | VideoChatGPT              | $19.17$%            | $18.09$%         |
> | VideoLLAMa                | $20.13$%            | $21.84$%         |
> | Video Chat2               | $20.88$%            | $21.95$%         |
> | LLaVA-Next-video          | $20.77$%            | $21.73$%         |
> | LLaVA-Next-video-dpo      | $20.34$%            | $21.41$%         |
>
> These results shows that the low accuracy of models is mostly because of poor visual recognition rather than knowing the technical skill names.
>
> 7. **Contradictory Observations**:  We believe that previous fine-grained actions recognition benchmarks such as Something something v2 [1] did not present a sufficient challenge for models as they lacked fine and intricate motions. In contrast, the Action Atlas benchmark presents a significant challenge, as the intricate motions in actions are so subtle that even with explicit descriptions of how to recognize these actions, models still struggle to visually identify them accurately.
>
> 8. **Human Baseline as Gold Standard**: We did human evaluation on Amazon Mechanical Turk with non-expert workers. For each question we provided the descriptions of each one of the options and instructed the workers to NOT use YouTube to answer any of the questions. The baseline accuracy of non-expert humans is $61.02$% which is close to $18$% higher than the $43.04$% accuracy of GPT-4o as the SOTA proprietary model.
>
>
> [1] Goyal, R., Ebrahimi Kahou, S., Michalski, V., Materzyńska, J., Westphal, S., Kim, H., Haenel, V., Fruend, I., Yianilos, P., Mueller-Freitag, M., Hoppe, F., Thurau, C., Bax, I., & Memisevic, R. (2017). The "something something" video database for learning and evaluating visual common sense. arXiv. https://arxiv.org/abs/1706.04261

---

> > ### Author Rebuttal · Authors · 2024-08-17
> >
> > 9. **Computational Complexity/Granularity**: Please refer to the human baseline results. Additionally, the consistent improvement observed when increasing the number of sampled frames indicates that recognizing actions in Action Atlas demands higher computational complexity.
> >
> > 10. **Claiming 1161 actions is a bit of a stretch**: To the best of our knowledge, Action Atlas is the first QA dataset for fine-grained action recognition. Previous fine-grained recognition datasets approached the task as a traditional classification problem with a fixed set of actions as classes. We deviated from that and followed datasets such as RefCOO [2] to use natural language to refer to individuals performing the actions which offers greater flexibility and more closely reflects how humans ask about actions.
> > Moreover, previous multiple-choice QA datasets have primarily focused on understanding coarse-grained actions and high level details of the video. We would appreciate it if the reviewer could provide further evidence supporting the inclusion of 200,000 actions in these options.
> >
> > [2] Yu, L., Poirson, P., Yang, S., Berg, A. C., & Berg, T. L. (2016). Modeling context in referring expressions. arXiv. https://arxiv.org/abs/1608.00272

---

> > > ### Comment · Area_Chair_CUJV · 2024-08-29
> > > **Reminder to response to author rebuttal**
> > >
> > > Dear Reviewer,
> > >
> > > The ddl for author and reviewer discussion is approaching. Please check the author rebuttal and leave some comments to respond to author rebuttal.
> > >
> > > Thanks,
> > > Your AC

---

> > > > ### Comment · Reviewer_jE6k · 2024-09-01
> > > >
> > > > Thank you to the authors for their responses. However, my concerns persist, as outlined below.
> > > >
> > > > **1. Poor dataset design-1.**  The proposed dataset contains answer options where more than one answer is correct. For example,
> > > >
> > > > Rebuttal Figure 1: Q: What best describes the action that the man wearing a gray sweatshirt and dark denim pants
> > > > performs?
> > > > 1. Slice shot 2. Flop shot 3. Chip shot 4. Double hit.
> > > >
> > > > In this case, action depicts both a Chip Shot (option 3) and Double Hit (option 4).
> > > >
> > > > GPT4o chooses option 3 and it is marked wrong, while in reality it is one of the correct answers.
> > > >
> > > > Due to the presence of multiple correct answers, even when models select a correct response, it is incorrectly judged as wrong. The paper misinterprets this flaw in the dataset design as an indication of poor model performance. In reality, the models are penalized even when their selections are correct, due to the dataset's poor construction. This issue renders the dataset ineffective for evaluating a model's ability to recognize fine-grained actions.
> > > >
> > > > To support multiple-choice questions effectively, the dataset should be designed so that each question has only one correct answer, a criterion that the current dataset fails to meet. As a result, the dataset, in its present form, is not suitable for reliable use.
> > > >
> > > > **2. Vagueness regarding the definition of fine-grained.** Unfortunately, neither the paper nor the rebuttal provided a definition of what paper considers as fine-grained actions, despite my specific request for clarification.
> > > >
> > > > **3. Poor dataset design 2.** Dataset contains many irrelevant factors that can give a false impression that the dataset is challenging. For example, dataset questions which involve identifying jersey numbers, which is not exactly a part of action. Jersey number identification is challenging, but not directly a part of fine grained nature of actions. Jersey numbers are not a property of fine grained actions, and as such the complexity stemming from jersey number identification cannot be used to claim the actions in the dataset are finegrained and challenging as also noted by another reviewer. Moreover, if jersey number identification is the goal, then there are datasets for jersey number identification out there.
> > > >
> > > > **4. Questionable claim that the dataset can test models’ capability to differentiate between 1161 actions (L87-89).** While I appreciate the rebuttal's response to this concern, I remain unconvinced by the reasoning provided. There is a clear distinction between answering a 4-choice question-answering and performing a 1161-way classification. Do the authors believe that the performance on their current 4-choice QA task would be equivalent to a 1161-way classification? If so, I would like to see evidence supporting this claim. In other words, the proposed dataset cannot replicate the level of difficulty inherent in a 1161-way classification (or differentiating between 1161 action choices) through their 4-choice QA format, as claimed in L87-89.
> > > >
> > > > While I value the time and effort invested in this work, I find it difficult to recommend this paper for acceptance.

---

> > > > > ### Author Response · Authors · 2024-09-02
> > > > > **Response to the very late comment submitted by the reviewer**
> > > > >
> > > > > Unfortunately, the comment is shown to be submitted on September 1st, after our rebuttal period had already closed. This prevents us from properly rebutting it or engaging in any helpful discussion with the other reviewers. We kindly ask for the AC's attention to this matter.
> > > > >
> > > > > Nevertheless, we address the reviewer’s concerns in this comment:
> > > > >
> > > > > 1. **Multiple correct actions**: Almost all questions in our dataset begin with the phrase “What _best_ describes the action …”. In the given golf video, the golfer hits the ball twice intentionally, which is noticeably different from a standard chip shot. Also in our human evaluation, humans have unanimously selected the double hit shot as what best describes the action. Furthermore, To show that GPT4 is actually not detecting a double hit which is obvious in the video, we provided the model with descriptions of the options and evaluated it on this sample using a chain-of-thought (CoT) approach. The output of GPT-4 when fed with 16 frames is as follows:
> > > > >
> > > > > “To find the action that best describes what the man in the gray sweatshirt is doing, let's compare it with the given options:\n\n1. **Slice shot**: This is a full swing where the ball curves from one side to another. It doesn't seem to fit because the man is close to a tree and not taking a full swing.\n2. **Flop shot**: Requires a steep, aggressive swing to get the ball high in the air and land softly. The man's setting and swing do not suggest this type of shot.\n3. **Chip shot**: A short, controlled swing to get the ball rolling on the green. The man's stance, position, and minimal swing could fit this description.\n4. **Double hit**: Involves hitting the ball more than once in a single swing. This action is usually accidental and results in a penalty. **The video doesn't clearly show him hitting the ball twice**.\n\nBy looking at the sequence:\n\n- The man's motion seems controlled and minimal, aiming to just tap the ball.\n- His stance close to the tree indicates a careful approach.\n- The swing is short and concise, which matches the description of a chip shot more than any other options.\n\nAfter this analysis, the best descriptor is:\n\n3”
> > > > >
> > > > > The issue does not go away even with feeding 32 total frames (the video lasts 4 second and sampling 32 frames would mean 8 frames per second):
> > > > >
> > > > > "The man in the video is getting ready to perform a golf shot. Let's go through the options one by one:\n\n1. **Slice shot:** This is a golf shot where the ball curves dramatically from left to right. The man is too close to a tree, and this setting doesn't seem favorable for a slice shot which typically requires an open clubface at impact or an outside-to-inside swing path.\n\n2. **Flop shot:** A high, soft shot used to clear an obstacle and land softly on the green, requiring an open clubface and a steep swing. This does not fit the current setup as the man's club doesn't seem to be particularly open, and the swing doesn't appear to be steep or aggressive enough for a flop shot.\n\n3. **Chip shot:** This is a short, low-trajectory shot used close to the green for getting the ball to roll towards the hole. The stance and the close proximity to the tree suggest that the man might be attempting a chip shot since he can make a short, controlled swing without much wrist action.\n\n4. **Double hit:** This occurs when the ball is struck more than once during a single swing, usually indicating a mistake or penalty. **The man initially strikes through the ball smoothly with controlled motion, making it improbable that this is a double hit**.\n\nUpon considering the options and given the golfer\u2019s stance, positioning near the tree, and the likely requirement to make a controlled, pendulum-like swing due to his limited movement space, it appears that the most fitting description is a chip shot.\n\nFinal choice:\n\n3"
> > > > >
> > > > > The model’s reasoning clearly shows that it has not detected a second hit which humans can easily recognize. As discussed in the paper, we believe that sparsely sampling frames in a video prevents models from capturing the exact moment when the subtle second hit happens.

---

> ### Author Response · Authors · 2024-08-26
> **Happy to address concerns**
>
> We wonder if we've addressed the reviewer's concerns and are happy to respond to any further questions.

---

> ### Author Response · Authors · 2024-09-02
> **Response to the very late comment submitted by the reviewer (part 2)**
>
> 2. **Vagueness regarding the definition of fine-grained.** Previous work such as MultiSports [1] specify a two level hierarchy for their fine-grained actions: First, they group the actions in a specific domain/sport into major categories (dribble, pass, shoot, etc.). The experts then list actions belonging to each category and iteratively refine them. However, we noticed that when humans compile these list, they often overlook some actions, particularly those belonging to the far tail of the distribution. The only difference in our approach is that, instead of relying on humans, we provided only few shot examples from the major categories to GPT-4 and ask the model to iteratively expand the list of actions belonging to those categories. That is the level of granularity in our work.
>
> 3. **Poor dataset design2**: When multiple individuals are in a video and each one of them is engaged in a different action, we need a way to identify and refer to the individual performing the action of interest. Previous datasets, like Multisports, have used bounding boxes for this purpose. However, we follow works like refcoco [2] and use natural language to refer to individuals. This aligns more closely with how humans naturally refer to an object or person and also better fits evaluating vision language models. To identify a specific individual, we ask annotators to use identifiers that _uniquely_ describes that person such as:
> - The name or number of jersey if visible.
> - Physical attributes such as hair color if it's unique to them.
> - Things that they do other than the given action (e.g. before or after it).
> - Combination of above.
>
> Therefore, there is nothing special about jersey numbers and it is just one of the means to refer to a specific individual when multiple people are in the video. In fact, in many instances the annotators used other identifiers such as clothing or hair color. In the specific example mentioned by the reviewer, the crowd-worker has naturally used the jersey number to identify the individual performing the diving header.
>
> 4. **Questionable claim that the dataset can test models’ capability to differentiate between 1161 actions**  We do not claim that our setup is equivalent to 1161 way classification. In fact, we would like to depart from that conventional n-way classification setup used in datasets like Something-something v2 [3] as we are evaluating foundation models with natural language capabilities. What we aim to argue in the paper is that correctly answering each question requires the model to recognize the best-fitting option given the question and the video. This involves differentiating between and recognizing all actions presented in the options for each question. Therefore, overall, our benchmark effectively tests the model’s recognition capability on all 1161 actions  (which stands at 1896 actions in the new version of the benchmark), not just the ground truth options. We will revise that sentence in question to better describe our objective.
>
> [1] Li, Y., Chen, L., He, R., Wang, Z., Wu, G., & Wang, L. (2021). MultiSports: A multi-person video dataset of spatio-temporally localized sports actions. arXiv. https://arxiv.org/abs/2105.07404
>
> [2] Yu, L., Poirson, P., Yang, S., Berg, A. C., & Berg, T. L. (2016). Modeling context in referring expressions. arXiv. https://arxiv.org/abs/1608.00272
>
> [3] Goyal, R., Ebrahimi Kahou, S., Michalski, V., Materzyńska, J., Westphal, S., Kim, H., Haenel, V., Fruend, I., Yianilos, P., Mueller-Freitag, M., Hoppe, F., Thurau, C., Bax, I., & Memisevic, R. (2017). The "something something" video database for learning and evaluating visual common sense. arXiv. https://arxiv.org/abs/1706.04261

---

### Official Review · Reviewer_Km21 · 2024-07-25

**Rating:** 6
**Confidence:** 4
**Correctness:** What is written seems to be correct.
**Clarity:** The paper is readable.

**Review:**

**Strengths:**

Challenging Dataset: The newly introduced video question answering dataset presents significant challenges, pushing the boundaries of current models' capabilities in fine-grained action recognition.

Innovative Data Selection Pipeline: The methodology for data selection is both novel and intriguing, providing a fresh approach to curating relevant video content.

**Weaknesses:**

Evaluation Methods: The evaluation focuses on methods based on Large Language Models (LLMs). It remains unclear how traditional deep learning methods would perform under similar conditions.

Scale of Benchmark: The proposed benchmark is relatively small-scale, consisting of only 554 videos. In contrast, existing datasets such as MSRVTT-QA, TGIF-QA, and DramaQA include over 10,000 videos with significantly more question-answer pairs.

Dataset Source Selection: The authors chose to source videos from YouTube despite the existence of established fine-grained sports action recognition datasets like MultiSports [1] and FineGYM [2]. The rationale behind this decision needs further justification.


[1] Multisports: A multi-person video dataset of spatio-temporally localized sports actions. CVPR 2021

[2] Finegym: A hierarchical video dataset for fine-grained action understanding. CVPR 2020

**Strengths:**

Please refer to **Review**.

**Additional Feedback:**

N.A.

**Documentation:**

The authors provided well-documented datasheets.

**Ethics:**

No ethical issue.

**Limitations:**

Please refer to **Review**.

**Opportunities For Improvement:**

The authors are suggested to evaluate the proposed dataset on the traditional deep-learning based method.

**Relation To Prior Work:**

The authors did a good job in the related work section.

**Summary And Contributions:**

The paper introduces ActionAtlas, a benchmark for evaluating video question answering (VideoQA) models on fine-grained action recognition tasks within short sports videos. ActionAtlas contains 554 videos spanning 284 actions across 42 sports, providing a challenging testbed for models to discern subtle differences in actions. The benchmark aims to push the limits of temporal and spatial localization in video understanding.

---

> ### Author Rebuttal · Authors · 2024-08-17
>
> We are glad that the reviewer believes that Action Atlas will push the boundaries of fine-grained action recognition and that its curation pipeline is innovative, novel and intriguing.
>
> We address the questions and weaknesses in what follows:
>
> 1. **Evaluating models with no LLMs:** We evaluated CLIP, a SOTA deep neural network with no LLMs, on our benchmark. To do so, we prefixed each option with the prompt “a video of“ and did a 4 or 5-way classification with the model. The following table shows the accuracy of CLIP ViT-L14-336 when varying the number of sampled frames which is slightly above random chance accuracy (random chance: 20.88%).
>
> | # sample frames | Acc.    |
> |-----------------|---------|
> | $1$             | $21.73$% |
> | $2$             | $21.73$% |
> | $4$             | $21.63$% |
> | $8$             | $22.38$% |
> | $16$            | $22.48$% |
>
> 2. **Scale of Benchmark**: We have nearly doubled the benchmark’s size, with the latest benchmark now including $934$ videos representing 580 distinct actions across $56$ sports. Unlike datasets such as MSRVTT-QA, TGIF-QA, and DramaQA, which consist of broad, general questions that often don't require multiple frames or specialized knowledge [1], our work focuses on evaluating  foundation models on fine-grained action recognition within specialized domains that have practical real-world applications. This type of data collection requires careful data curation and rigorous quality control. In this context, a better comparison for our dataset would be something like GPQA [2], which includes $500$ samples for evaluating language models on graduate level questions across different fields.
> Furthermore, the majority of data in datasets like MSRVTT-QA, TGIF-QA, and DramaQA is in curated training sets. We argue that foundation models should acquire domain knowledge, including fine-grained action recognition, through pretraining on internet-scale multimodal datasets, as emphasized by modern benchmarks like MMLU [3]–see the section 'The Internet as a Training Set' in the MMLU paper. As outlined in section 3.2 of our paper, we have compiled a list of over $8$ million videos relevant to the fine-grained actions featured in our benchmark. These videos along with their titles, descriptions, and ASR transcripts can be used as valuable sources for pretraining a video model. We will also be releasing the ids of these videos alongside the benchmark.
>
> 3. **Dataset Source Selection**: Multisports dataset includes only 66 actions, leaving many sports-specific actions and moves uncovered, which we easily discovered by prompting GPT-4. For instance, for in soccer, Multisports contain $15$ actions, while our dataset includes $73$ actions covering fine-grained dribbles, shots, and passes. Finegym is limited to gymnastics moves, excluding other sports entirely. These limitations motivated us to source new videos from YouTube. With our scalable pipeline in place in the current version, we plan to expand the dataset to include more actions across a wide range of sports and potentially other domains (e.g. arts and crafts, dancing, medical actions etc.) in our future work.
>
>
> [1] Buch, S., Eyzaguirre, C., Gaidon, A., Wu, J., Fei-Fei, L., & Niebles, J. C. (2022). Revisiting the "video" in video-language understanding. arXiv. https://arxiv.org/abs/2206.01720
>
> [2] Rein, D., Hou, B. L., Stickland, A. C., Petty, J., Pang, R. Y., Dirani, J., Michael, J., & Bowman, S. R. (2023). GPQA: A graduate-level Google-proof Q&A benchmark. arXiv. https://arxiv.org/abs/2311.12022
>
> [3] Hendrycks, D., Burns, C., Basart, S., Zou, A., Mazeika, M., Song, D., & Steinhardt, J. (2021). Measuring massive multitask language understanding. arXiv. https://arxiv.org/abs/2009.03300

---

> > ### Author Response · Authors · 2024-08-26
> > **Happy to address concerns**
> >
> > We wonder if we've addressed the reviewer's concerns and are happy to respond to any further questions.

---

> > > ### Comment · Reviewer_Km21 · 2024-08-27
> > > **Response to the rebuttal**
> > >
> > > Apologies for the delayed response. The authors have addressed my main concerns, so I am updating my rating to 6: Marginally above the acceptance threshold.

---

> ### Author Response · Authors · 2024-08-22
> **Happy to address concerns**
>
> We wonder if we've addressed the reviewer's concerns and are happy to respond to any further questions.

---

### Official Review · Reviewer_5HLL · 2024-07-25
**ActionAtlas a new Benchmark for Video understanding**

**Rating:** 8
**Confidence:** 3
**Clarity:** yes with further details given in a s…

**Review:**

This is a well-written paper with a clear motivation. The authors went into great detail to describe the dataset gathering and curation and filtering process. They had a multi-step data collection and filtering pipeline which is explained nicely and additional information is provided in supplementary.
My concern is, given that the authors used YouTube sports videos for the generation of the data, whether it might lead to copyright infringement from the owner of the videos.
Can the same video be tagged for multiple different action prompts? each video is <=30 Sec long. In that 30 sec, multiple fine-grained actions can be taken in fast-paced sports like football. If yes, then will tagging the same video to multiple different actions cause any learning disability or confusion to the video recognition model? If not then why do the authors think so?
The authors have provided a detailed benchmark on various open sourced and proprietory models and provided interesting observations that even though our current SOTA models do well on simple action recognition tasks, they still fail to identify and infer actions when the details for identifying the action are spread spatially as well as temporally. This is the case in sports or cooking. Our current models focus on recognising action from a single image frame which is highlighted from the benchmarking results from this dataset. Thus providing future scope and direction for a better understanding of video data.

The authors have provided the dataset in google drive and promised to host the data on Hugging face and via a git hub link in future.

**Strengths:**

Well written paper with clear problem statement regarding video action recognition task that this dataset is meant to address.

The dataset collection and filtering pipeline is explained in details with proper diagrams.

They have conducted a detailed Benchmark study on the proposed dataset with results that highlight the shortcomings of current SOTA models and opens up avenues for future research in video understanding.

Dataset is available in google drive and promised to host the data in hugging face and via a git hub link in future.

**Additional Feedback:**

None

**Correctness:**

Yes this is a dataset papar with detailed benchmarking results . To my knowledge and understanding the data construction pipeline presented by the authors seem sound. The reasoning  behind the design of the experiments are properly cited and seems correct to my knowledge.

**Documentation:**

Documentation is decently done. However, currently the dataset is being hosted in a Google drive, however the authors have promised to host the data in Hugging face and provide the github link.

**Ethics:**

This dataset is curated from Youtube videos and I am not sure whether this might lead to any possible copyright infringement from the owners or not.

**Limitations:**

With better understanding of video models, we will be able to generate videos of better accuracies and spatio-temporal consistencies, which can lead to abuse and misinformation campaigns that would be difficult to differentiate from a real video.

This dataset is curated from Youtube videos and I am not sure whether this might lead to any possible copyright infringement from the owners or not.

For further details please refer to my review.

**Opportunities For Improvement:**

Please refer to my details comments in Review.

**Relation To Prior Work:**

yes it is clearly mentioned with benchmarking results to support the claims.

**Summary And Contributions:**

The authors introduced ActionAtlas a VideoQA benchmark for evaluating VLMs on fine-grained real-world action recognition. They have created a new VideoQA dataset for fine-grained action recognition in regarding sports data that forces the VLMs to  correctly localize the action, both temporally and spatially. They have done an extensive benchmarking of SOTA models both open-sourced and Proprietory on the dataset and presented interesting observations that offer many future avenue of research

---

> ### Author Rebuttal · Authors · 2024-08-17
>
> We appreciate the reviewer's recognition of the detailed explanation of the annotation pipeline, supported by clear diagrams. We're also pleased that they believe the benchmark shows the weaknesses of current SOTA models in action understanding and opens up avenues for future research in video understanding.
>
> We address the questions and weaknesses in what follows:
>
> 1. **Copyright infringement**: As highlighted by the ethics reviewer, we will take the necessary steps for copyrighted videos, including preserving the metadata.
>
> 2. **Can the same video be tagged for multiple different action prompts?** Yes, many of the 30-second segments contained multiple actions, which is why we asked crowd-workers to further refine the localization of the fine-grained actions of interest. For future work, we plan to involve domain experts to annotate all the actions occurring in the videos we collect, using these annotations as the ground truth captions. This approach could make the benchmark more challenging and engaging, as it would also test models' abilities to understand long-form content.
>
> 3. **If yes, then will tagging the same video to multiple different actions cause any learning disability or confusion to the video recognition model?** Our results show that current VLMs struggle to accurately recognize a single fine-grained action in a short video. This suggests that in longer videos with multiple actions, these recognition failures may accumulate, leading to even poorer overall performance.

---

> > ### Comment · Area_Chair_CUJV · 2024-08-29
> > **Reminder to response to author rebuttal**
> >
> > Dear Reviewer,
> >
> > The ddl for author and reviewer discussion is approaching. Please check the author rebuttal and leave some comments to respond to author rebuttal.
> >
> > Thanks,
> >
> > Your AC

---

> > ### Comment · Reviewer_5HLL · 2024-08-29
> > **Response to rebuttal**
> >
> > I have gone through the rebuttal by the authors and they have addressed my minor concerns regarding the paper and updated my final review score for the paper.

---

### Author Rebuttal · Authors · 2024-08-17

We thank the reviewers for their valuable feedback. We are glad that reviewer Km21 recognized that Action Atlas will push the boundaries of fine-grained action recognition, and that its curation pipeline is innovative, novel and intriguing. Reviewer DiB7 has also highlighted the novelty of the work, particularly the semi-automatic collection pipeline and the use of GPT4-text for faster identification of candidate temporal segments (Figure 3 of the submission), and the importance of analysis on the number of input frames. Reviewer 5HLL has acknowledged the detailed explanation of the annotation pipeline with proper diagrams (Figure 2 of the submission), and noted that the benchmark reveals the weaknesses of current SOTA models in action understanding which opens up avenues for future research in video understanding.

We would like to provide some updates regarding the benchmark and the evaluations:

1. We have nearly doubled the benchmark’s size, with the latest benchmark now including $934$ videos representing $580$ distinct actions across $56$ sports. The first and second table at the bottom shows the results of proprietary and open multi-modal large language models on the new benchmark.

2. We conducted a human evaluation with crowd-workers on Amazon Mechanical Turk to get non-expert human’s performance as a baseline. To do so, we prompted GPT4-o to provide a 2-3 sentence description of each of the actions. We then provided these descriptions along with the options and asked workers to answer the questions. We instructed the workers to **NOT** use YouTube when answering the question but they were allowed to do text search on Google and using AI chatbots provided they don’t use the vision capability of the chatbot. The accuracy of workers was $61.02$% which is close to $18$% higher than the performance of GPT-4o as the best AI model.

3. We conducted qualitative analysis and provided five examples of GPT-4o’s failures with 16 frames inputs as the SOTA on our task in the uploaded rebuttal PDF. For this analysis, we prompted the model to explain its reasoning behind choosing a specific option. We found that most failures were because of hallucinations about actions that did not actually happen in the video. Specifically, in Figures 2 and 4 of the PDF, the model overlooked key movements in the ground truth actions and their critical landmarks—such as the body's rotation in the "rose" move in Figure 2 and the brief pause in the "hesitation" move in Figure 4—while instead hallucinating actions that did not happen (e.g., "hooking the heel" in Figure 2 and "spinning" in Figure 4). Additionally, the model struggled to recognize finer subcomponents of certain actions. For instance, in Figure 1, it failed to notice that the person hit the golf ball a second time. In rare cases, like in Figure 5, the model hallucinated the action occurring in the video and reasoned about this hallucination but ultimately chose a totally different option. In these instances, there were errors both in visual recognition and in reasoning. We will include these examples, along with potentially more, in the paper.

Proprietary Multi-modal Large Language Models results
| Model             | # Input frames | Accuracy    |
|--------------------------|--------------------------|--------------------------|
| **Random chance**  | -                 | $20.88$%  |
| **Non-expert Human**                 | Full video   | $61.02$%  |
| **GPT-4o**                | $1$                 | $33.08$%  |
|                                  | $2$                 | $31.47$%  |
|                                  | $4$                 | $39.5$%   |
|                                  | $8$                 | $41.54$%  |
|                                  | $16$             | $43.04$%  |
|                                  | $32$             | $41.43$%  |
|  **Gemini Pro 1.5**   | Video mode              | $33.83$%  |
|                   | $2$                              | $29.12$%  |
|                   | $4$                              | $30.72$%  |
|                   | $8$                              | $29.97$%  |
|                   | $16$                            | $32.44$%  |
|                   | $32$                            | $32.44$%  |
| **GPT-4o mini**                |$1$                 | $30.19$%  |
|                                  | $2$                 | $27.83$%  |
|                                  | $4$                 | $33.40$%   |
|                                  | $8$                 | $30.19$%  |
|                                  | $16$             | $32.22$%  |
|                                  | $32$             | $31.15$%  |

Open models:
| Model                     | # input frames | Acc.    |
|---------------------------|----------------|---------|
| VideoChatGPT              | $16$           | $19.17$% |
| Video Llama               | $16$           | $20.13$% |
| Video Lavit               | $24$           | $19.38$% |
| VideoChat2                | $16$           | $20.88$% |
|                           | $32$           | $20.88$% |
|                           | $64$           | $21.31$% |
| LLaVA-Next-video          | $16$           | $20.77$% |
|                           | $32$           | $21.09$% |
|                           | $64$           | $22.91$% |
| LLaVA-Next-video-dpo      | $16$           | $20.24$% |
|                           | $32$           | $20.99$% |
|                           | $64$           | $23.34$% |

---

> ### Author Rebuttal · Authors · 2024-08-17
>
> 4. To make sure that the poor performance of AI models is not because of lack of knowledge about the fine-grained actions or not knowing their names, similar to the human study, we provided descriptions of actions alongside the actions. We observed marginal improvements, as shown in the tables for both GPT-4o and open models when provided with 16 frames. This indicates that the low performance is primarily due to poor visual recognition rather than language understanding.
>
>
> | Model | Acc. without description | Acc. with description        |
> |-----------------|--------------------------|------------------------------|
> | GPT-4o            | $43.04$%                 | $44.00$% (+$0.96$%)          |
>
>
> Open models:
> | Model                     | Acc. Without description | Acc. With description |
> |---------------------------|---------------------|------------------|
> | mPlug-owl                 | $19.49$%            | $20.87$%         |
> | VideoChatGPT              | $19.17$%            | $18.09$%         |
> | VideoLLAMa                | $20.13$%            | $21.84$%         |
> | Video Chat2               | $20.88$%            | $21.95$%         |
> | LLaVA-Next-video          | $20.77$%            | $21.73$%         |
> | LLaVA-Next-video-dpo      | $20.34$%            | $21.41$%         |
>
>
> 5. We evaluated OpenAI's CLIP, a SOTA deep neural network with no LLMs, on our benchmark. To do so, we prefixed each option with the prompt “a video of“ and did a 4 or 5-way classification with the model. The following table shows the accuracy of CLIP ViT-L14-336 when varying the number of sampled frames which are all slightly above random chance accuracy (random chance: $20.88$%).
>
> | # sample frames | Acc.    |
> |-----------------|---------|
> | $1$             | $21.73$% |
> | $2$             | $21.73$% |
> | $4$             | $21.63$% |
> | $8$             | $22.38$% |
> | $16$            | $22.48$% |

---

### Decision · Program_Chairs · 2024-09-26

**Decision:**

Accept (Poster)

**Comment:**

This paper receives mixed reviews: three positive ratings and one negative rating. In general, the reviewers appreciate this paper proposes a new benchmark related to fine-grained actions that is valuable for evaluating the MLLM on the task of action understanding. One reviewer raises some concern on the detailed design of building the benchmark. After a careful check of paper, reviewer comments, and author rebuttal, the AC thinks the major concern is well addressed by the authors and agrees with the majority of the reviewers. Thus, the AC makes an accept recommendation.

In addition, the AC agrees that the term "fine-grained action recognition" is indeed misleading, as the benchmark is different to traditional action recognition benchmarks. Thus, it is suggested to revise the term to "fine-grained actions". This paper also misses a very relevant reference on fine-grained action recognition in sports, which is also discussed in the rebuttal:

MultiSports: A Multi-Person Video Dataset of Spatio-Temporally Localized Sports Actions [ICCV 2021]

Please cite and discuss this work in your final version.